# Re-focusing visual working memory during expected and unexpected memory tests

**Sisi Wang\*, Freek van Ede\***

Institute for Brain and Behavior Amsterdam, Department of Experimental and Applied Psychology, Vrije Universiteit Amsterdam, Amsterdam, Netherlands

## eLife Assessment

This **important** study provides significant insights into the dynamics of attentional re-orienting within visual working memory, demonstrating how expected and unexpected memory tests influence attention focus and re-focus. The evidence supporting these conclusions is **convincing**, with the use of state-of-the-art methodologies. This work will be of interest to cognitive neuroscientists studying attention and memory.

**Abstract** A classic distinction from the domain of external attention is that between anticipatory orienting and subsequent re-orienting of attention to unexpected events. Whether and how humans also re-orient attention 'in mind' following expected and unexpected working-memory tests remains elusive. We leveraged spatial modulations in neural activity and gaze to isolate re-orienting within the spatial layout of visual working memory following central memory tests of certain, expected, or unexpected mnemonic content. Besides internal orienting after predictive cues, we unveil a second stage of internal attentional deployment following both expected and unexpected memory tests. Following expected tests, internal attentional deployment was not contingent on prior orienting, suggesting an additional verification – 'double checking' – in memory. Following unexpected tests, re-focusing of alternative memory content was prolonged. This brings attentional re-orienting to the domain of working memory and underscores how memory tests can invoke either a verification or a revision of our internal focus.

**\*For correspondence:**
s.wang5@vu.nl (SW);
freek.van.ede@vu.nl (FvE)

**Competing interest:** The authors declare that no competing interests exist.

## Introduction

Attention enables us to select and prioritize relevant information and can be directed not only to external sensations but also to internal representations held in working memory (*Griffin and Nobre, 2003*; *Panichello and Buschman, 2021*; *Souza and Oberauer, 2016*; *van Ede and Nobre, 2023*). In everyday life, attention is often directed to sensory and mnemonic information that is likely to become relevant based on probabilistic cues, but new events may also prompt us to revise our attentional focus, particularly when these events come unexpected.

This underscores a classic distinction between two separate attentional stages: attentional orienting and re-orienting. For example, following a probabilistic pre-cue, participants may initially orient their attention to the left where a visual target is expected, but if the target unexpectedly comes on the right, attention requires to be re-oriented to the right (*Posner, 1980*).

Studies on external attention have extensively characterized the mechanisms of both orienting of attention following predictive probabilistic cues and re-orienting of attention following unexpected perceptual targets (*Carrasco, 2018*; *Corbetta et al., 2002*; *Corbetta et al., 2000*; *Doricchi et al.,*

*2010*; *Posner, 1980*; *Posner et al., 1982*; *Posner and Cohen, 1984*). In contrast, studies on internal attention – directed to representations held in working memory – have almost exclusively focused on just the former: the mechanisms supporting initial orienting following predictive retrocues (for reviews, see *Myers et al., 2017*; *Oberauer, 2019*; *Sahan et al., 2020*; *Souza and Oberauer, 2016*; *van Ede and Nobre, 2023*) or temporal expectations (*van Ede et al., 2017*; *Zokaei et al., 2019*). These studies have made clear, among others, how the brain relies on memorized locations to orient to specific objects held within the spatial layout of working memory (*de Vries et al., 2023*; *Draschkow et al., 2022*; *Fu et al., 2022*; *Günseli et al., 2019*; *Kuo et al., 2009*; *Lepsien et al., 2005*; *Li et al., 2023*; *Liu et al., 2022*; *Macedo-Pascual et al., 2022*; *Myers et al., 2015*; *Poch et al., 2017*; *Poch et al., 2014*; *Schneider et al., 2015*; *van Ede, 2018*; *van Ede et al., 2021*; *van Ede et al., 2020*; *van Ede et al., 2019a*; *van Ede et al., 2019b*; *Wallis et al., 2015*; *Wolff et al., 2017*). Yet, whether deployment of spatial attention within visual working memory also occurs in response to ensuing memory tests – and how this depends on whether working memory tests prompt mnemonic content that is certain, expected, or unexpected to become tested – has remained elusive.

To fill this gap, we designed a visual working memory task in which we varied the reliability by which retrocues predicted the upcoming working memory test. Retrocues were either 100% reliable, 80% reliable, or 60% reliable, yielding memory tests for items that were certain (100%), expected (80/60% reliable and valid), or unexpected (80/60% reliable but invalid) to be probed for report. Building on ample prior studies, we tracked spatial orienting of attention within the spatial layout of visual working memory using two established markers: the lateralization of 8–12 Hz posterior EEG-alpha activity (*Fu et al., 2022*; *Günseli et al., 2019*; *Li et al., 2023*; *Macedo-Pascual et al., 2022*; *Myers et al., 2015*; *Poch et al., 2017*; *Poch et al., 2014*; *Schneider et al., 2016*; *Schneider et al., 2015*; *van Ede, 2018*; *van Ede et al., 2019a*; *Wallis et al., 2015*; *Wolff et al., 2017*) and the spatial biasing of (micro) saccades (as in *de Vries et al., 2023*; *Draschkow et al., 2022*; *Liu et al., 2022*; *van Ede et al., 2021*; *van Ede et al., 2020*; *van Ede et al., 2019a*). By presenting memory items laterally, but memory tests centrally, we could use these spatial markers to test whether attention also re-orients inside the mind when specific memory content becomes tested – and to assess how such re-orienting depends on prior expectations regarding the to-be-tested memory content.

To preview our results, we first confirm that our two spatial markers track orienting of attention in response to the retrocue, in a manner that scales with cue reliability (cf. *Fu et al., 2022*; *Gould et al.,*

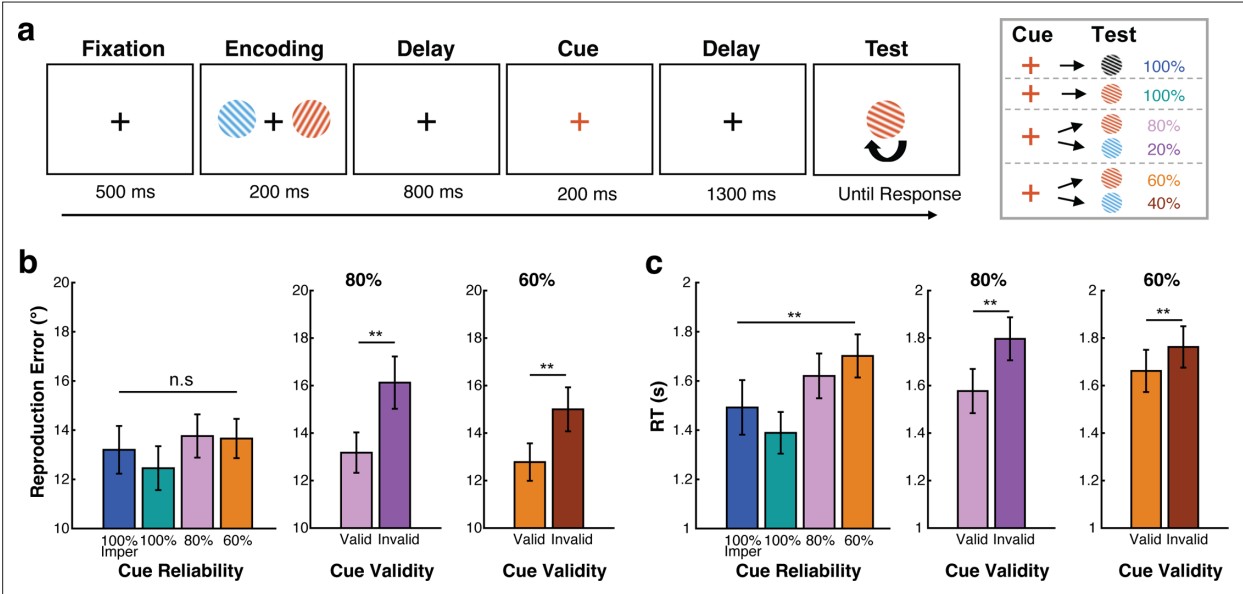

**Figure 1.** Better performance for expected vs. unexpected working memory tests. (**a**) Task schematic. Participants (n=24) memorized two visual items, of which one would become tested for orientation reproduction. During the delay, a retrocue would indicate which memory item would most likely be tested. Across blocks, the retrocue was either 100%, 80%, or 60% reliable, or 100% reliable and imperative (no repeat of color at test) as illustrated in the right panel. (**b**) Continuous reproduction error across four cue-reliability conditions (left panel), and for valid and invalid-cue trials in the 80% and 60% cue-reliability blocks (right panels). (**c**) Response time across conditions. Error bars represent ± 1 SEM. *, **, n.s represent significance level $p<0.05$, $p<0.01$, and non-significant after Bonferroni correction.

*2011*; *Günseli et al., 2019*; *Haegens et al., 2012*). Having established the sensitivity of our markers and the use of the retrocue, we next uncover evidence for spatial re-orienting within the mind in response to the central memory test – including following tests of expected memory objects, but only when the to-be-tested memory object cannot be anticipated with complete certainty. Moreover, when the memory test prompts selection of the unexpected memory content, we unveil how re-orienting in mind does not take off later, but does last longer, consistent with slower responses following unexpected (invalidly cued) memory tests.

## Results

Human participants performed a visual working memory task in which retrocues directed attention to either of two visual items (colored oriented gratings) held in working memory (*Figure 1a*). Across different blocks, cues were either 100% reliable and imperative (no repeat of cue color at test), 100% reliable but non-imperative (cue color repeated at test), 80% reliable, or 60% reliable as to whether they correctly predicted the to-be-tested memory item (as illustrated to the right in *Figure 1a*).

Our focus was on delineating the dynamic attentional processes governing effective memory-guided behavior. To this end, we turned to EEG-alpha and gaze signatures of internal attentional orienting within the spatial layout of visual working memory. Critically, we tracked orienting of attention not only in response to the retrocue (as in ample prior studies, e.g. *Fu et al., 2022*; *Günseli et al., 2019*; *Li et al., 2023*; *Macedo-Pascual et al., 2022*; *Myers et al., 2015*; *Poch et al., 2017*; *Poch et al., 2014*; *Schneider et al., 2016*; *Schneider et al., 2015*; *van Ede, 2018*; *van Ede et al., 2019a*; *Wallis et al., 2015*; *Wolff et al., 2017*), but also in response to the memory test, as a function of whether the memory test was (a) certain (after 100% reliable cues), (b) uncertain but conform the cue-induced expectation (after 80/60% *valid* cues), or (c) different than expected (after 80/60% *invalid* cues). To do so, we leveraged a unique aspect of our task in which we always presented the test stimulus (i.e. the memory probe) centrally. This enabled us to track re-orienting to the left/right memory item, and disentangle such re-orienting 'in mind' from visual processing of the test stimulus itself.

In what follows, we first present our behavioral performance findings to confirm that participants used the cue to benefit memory-guided behavior. We then turn to our EEG and gaze signatures of attentional orienting during working memory following the retrocue. Finally, we turn to our critical findings of attentional re-orienting following the memory test at the end of the working memory delay.

### Better performance following expected vs. unexpected memory tests

Because we varied the reliability of the cue across blocks, we first evaluated reproduction errors and reaction times (RT) across block types (*Figure 1b and c*). We found comparable accuracy (reproduction errors) across the four block types ($F_{(3,69)} = 1.378$, $p=0.257$, $\eta^2=0.057$), while for RTs we found a significant main effect of block type ($F_{(3,69)} = 8.994$, $p<0.001$, $\eta^2=0.281$). This was corroborated by a significant linear trend ($F_{(1,23)} = 15.304$, $p=0.001$, $\eta^2=0.400$), with faster responses in blocks with more reliable cues.

To more directly quantify cueing effects on performance, we turned to the blocks with 80% and 60% reliable cues, as these contained memory tests that were either expected (following a valid retrocue) or unexpected (following an invalid retrocue). This confirmed clear behavioral consequences of cue-induced expectations on memory-guided behavioral performance, with higher reproduction accuracy (lower errors) and faster response speed (lower RTs) when the memory test probed the expected (validly cued) vs. the unexpected (invalidly cued) memory item (80% reliable-cue blocks, accuracy: $t_{(23)} = 5.357$, $p<0.001$, RT: $t_{(23)} = -5.651$, $p<0.001$; 60% reliable-cue blocks, accuracy: $t_{(23)} = 3.850$, $p=0.012$, RT: $t_{(23)} = -5.286$, $p<0.001$). As can be seen in *Figure 1b and c*, effects of cue validity were also larger following 80% (2.948 degrees and 219 ms better after valid vs. invalid cues) vs. 60% (2.221 degrees and 100 ms better after valid vs. invalid cues) reliable cues. For RT, this was corroborated by a significantly larger cue-validity effect following 80% vs 60% reliable cues ($t_{(23)} = -3.497$, $p=0.002$), though for accuracy this difference did not reach significance ($t_{(23)} = -1.533$, $p=0.139$). Together, these results show that participants used the cue even when the cue was 80% or 60% reliable, and did so in a graded manner that depended on cue reliability.

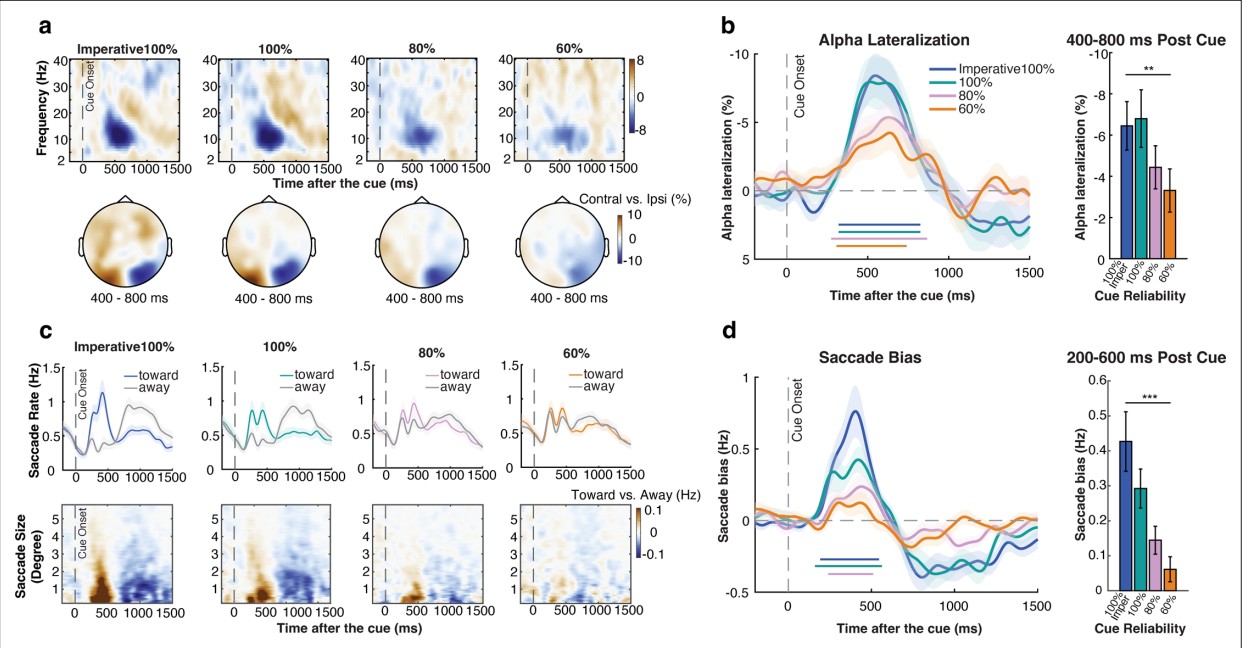

**Figure 2.** Attentional orienting after the cue shows graded spatial modulations in both EEG-alpha activity and in gaze. (**a**) Top: time-frequency spectrum (contralateral minus ipsilateral) across frequency bands (2–40 Hz) following the cue (n=23). Bottom: topographic distribution of 8–12 Hz EEG-alpha lateralization across conditions (averaged across 400–800 ms after the cue). (**b**) Left panel: time-series of alpha lateralization across conditions. Right panel: averages of alpha lateralization in a pre-defined time window (400–800 ms after cue, as in *van Ede et al., 2019a*) across conditions. (**c**) Top: saccade rates separately for toward and away saccades across cue conditions relative to cue onset (n=22), colored lines represent saccades toward the memorized location of the cued item while gray lines represent saccades away from the memorized location of the cued item. Bottom: spatial saccade bias (toward minus away, color-coded) as a function of saccade size. The difference between toward and away saccades (with red colors denoting more toward saccades) was predominantly driven by saccades in the micro-saccade range (<2° visual degree) rather than looking back to the original location of the items that were centered at 6 degrees during encoding. (**d**) Left panel: spatial saccade bias (toward minus away) across conditions. Right panel: averages of saccade bias in a pre-defined time window (200–600 ms after cue as in *Liu et al., 2022*) across conditions. Colored horizontal lines above the x-axes in b,d indicate significant clusters (*p*<0.05). Shading and error bars represent ± 1 SEM. *, **, *** represent significance level *p*<0.05, *p*<0.01, and *p*<0.001.

The online version of this article includes the following figure supplement(s) for figure 2:

**Figure supplement 1.** Spatial saccade biases associated with initial orienting after the retrocue and re-orienting after the memory test are preserved when exclusively considering fixational 'micro' saccades <2 degrees.

## Attentional orienting after the cue shows graded spatial modulations in both EEG-alpha activity and in gaze

Having established robust cueing benefits on memory-guided behavioral performance, we next aimed to track the dynamics of cue-driven attentional orienting, as a function of cue reliability. We here uniquely did so for two separate markers that have each been implicated in attentional orienting within the spatial layout of visual working memory: the lateralization of 8–12 Hz EEG-alpha activity (e.g. *Fu et al., 2022*; *Günseli et al., 2019*; *Li et al., 2023*; *Macedo-Pascual et al., 2022*; *Myers et al., 2015*; *Poch et al., 2017*; *Poch et al., 2014*; *Schneider et al., 2016*; *Schneider et al., 2015*; *van Ede, 2018*; *van Ede et al., 2019a*; *Wallis et al., 2015*; *Wolff et al., 2017*), and the spatial biasing of saccade directions (e.g. *de Vries et al., 2023*; *Draschkow et al., 2022*; *Liu et al., 2022*; *van Ede et al., 2021*; *van Ede et al., 2020*; *van Ede et al., 2019a*).

We first considered alpha lateralization. In each of our four cue-reliability conditions, we found a clear alpha lateralization (cluster *p*: 100% imperative: *p*<0.001; 100%: *p*<0.001; 80%: *p*<0.001; 60%: *p*=0.004), with a characteristic posterior topography (*Figure 2a*). Consistent with prior work (*Fu et al., 2022*; *Gould et al., 2011*; *Günseli et al., 2019*; *Haegens et al., 2012*), the spatial alpha lateralization was graded as function of cue reliability, being most pronounced for more reliable cues (*Figure 2b*). To quantify this statistically, we extracted alpha lateralization in the a-priori defined time window from 400 to 800 ms post cue (based on *van Ede et al., 2019a*), as shown in *Figure 2b*. A repeated measures

ANOVA revealed a significant main effect of cue reliability ($F(3,66) = 4.711$, $p=0.005$, $\eta^2=0.176$), and a significant linear trend ($F(1,22) = 10.725$, $p=0.003$, $\eta^2=0.328$), with stronger alpha lateralization following more reliable cues.

In addition to our EEG-alpha marker of spatial orienting in visual working memory, we have recently uncovered and reported how attentional shifts to visual objects in working memory can also be tracked by reliable spatial biases in gaze (see *de Vries et al., 2023*; *Draschkow et al., 2022*; *Liu et al., 2022*; *van Ede et al., 2021*; *van Ede et al., 2020*; *van Ede et al., 2019a*; for complementary findings see also *Engbert and Kliegl, 2003*; *Hafed and Clark, 2002*; *Johansson and Johansson, 2014*; *Richardson and Spivey, 2000*; *Spivey and Geng, 2001*; *Wynn et al., 2019*), driven predominantly by microsaccades (see: *de Vries et al., 2023*; *Liu et al., 2023*; *Liu et al., 2022*). To date, however, we have only ever shown this following retrocues that were 100% reliable. The current experiment enabled us to assess whether, like EEG-alpha lateralization, this gaze signature of internal attentional orienting also showed a graded modulation as a function of cue reliability.

*Figure 2c* shows saccade rates following the cue, separately for saccades that occurred toward the focused memory item vs. in the opposite, away direction. Similar to our EEG-alpha marker, this saccadic marker of internal attention shifts was also most pronounced when the cue was 100% reliable, though qualitatively similar effects were observed following 80% and 60% reliable cues *Figure 2c, d*; cluster *p*: 100% imperative: $p=0.002$; 100%: $p=0.002$; 80%: $p=0.002$; 60%: $p=0.341$.

As in prior studies from our lab with similar experimental set-ups, internal attentional focusing was predominantly driven by fixational micro-saccades (small, involuntary eye movements around current fixation). To reveal this in the current study, we decomposed and visualized the observed saccade-rate effect as a function of saccade size (*Figure 2c*), following the same procedure as we have adopted in other recent studies on this bias (*de Vries et al., 2023*; *Liu et al., 2023*; *Liu et al., 2022*). As shown in the saccade-size-over-time plots in *Figure 2c*, also in the current study, the difference between toward and away saccades (with red colors denoting more toward saccades) was predominantly driven by fixational saccades in the micro-saccades range (<2°).

Moreover, as shown in *Figure 2—figure supplement 1a* complementary analyses show that our time course (saccade bias) results hold even when exclusively considering eye movements below 2 visual degrees that we defined as 'fixational' provided that the memory items were presented 6 visual degrees from the fixation during encoding. This further corroborates that the bias observed during internal attentional focusing was predominantly driven by fixational micro-saccades rather than looking back to the encoded location of the memory items (*Johansson and Johansson, 2014*; *Richardson and Spivey, 2000*; *Spivey and Geng, 2001*; *Wynn et al., 2019*).

To extract this bias, we subtracted toward and away saccades (*Figure 2d*), and zoomed in on the a-priori-defined window from 200 to 600 ms after the cue (based on *Liu et al., 2022*). The saccade bias was also graded as a function of cue reliability, as confirmed by a significant main effect of cue reliability ($F(3,63) = 11.049$, $p<0.001$, $\eta^2=0.345$), and a significant linear trend ($F(1,21) = 18.695$, $p<0.001$, $\eta^2=0.471$), with a more pronounced spatial saccade bias when the cue was more reliable.

Together, our two markers of spatial shifts of attention in visual working memory thus each show a graded modulation by cue reliability, with larger modulations following more reliable cues. It's worth noting that while alpha lateralization shows very comparable amplitudes in the imperative-100% and 100% conditions, the saccade bias was larger following imperative-100% vs. 100% reliable cues. This may reflect a difference between these two cueing conditions that is selectively picked up by our gaze marker (though it may also reflect differential sensitivity of our two markers to different sources of noise). While the two markers each had their own characteristic patterns, a direct comparison between the two markers is non-trivial (as they are two distinct dependent variables) and was beyond the scope of the current study (for studies targeting the inter-relation between these two markers, we refer the reader to *Liu et al., 2023*; *Liu et al., 2022*).

## Signatures of attentional orienting also reveal attentional re-orienting after the memory test, but only when the test is not certain

A unique feature of our task was that the two memory items were always left and right at encoding, while the working memory test was always central. This enabled us to investigate spatial orienting of attention not only in response to the retrocue (as described above), but also in response to the test stimulus that prompted a memory-guided report. Any lateralized modulation in response to the

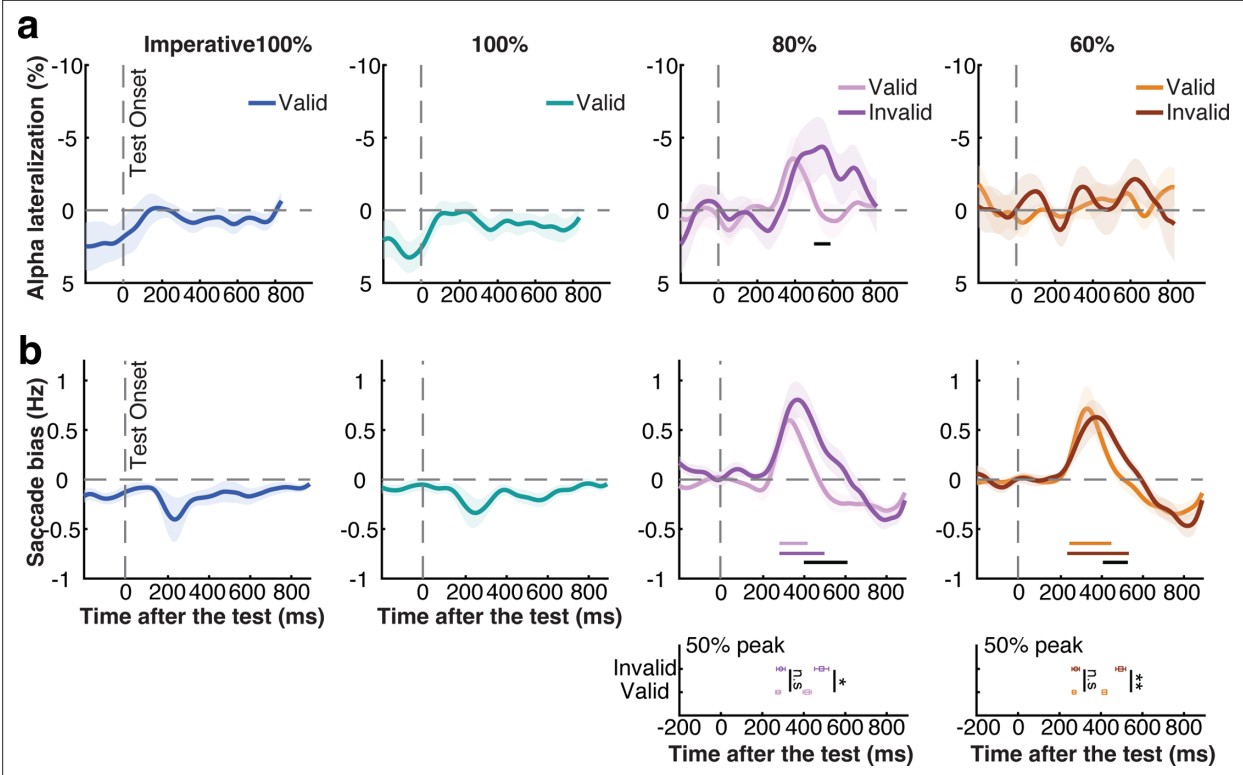

**Figure 3.** Attentional re-orienting after the memory test. (**a**) Alpha lateralization towards the memorized location of the tested item across conditions, here relative to after the onset of the memory test instead of the cue. From left to right panels, the alpha lateralization across imperative 100%, informative 100%, 80%, and 60% reliable-cue conditions. (**b**) Saccade bias towards the memorized location of the tested item across conditions after the onset of the memory test. Colored horizontal lines above x-axis indicate significant clusters (*p*<0.05). The black horizontal lines indicate significant clusters (*p*<0.05) comparing validly and invalidly cued tests. Plots at the bottom of panel **b**: onset and offset latency defined by the 50% of the peak of the spatial saccade bias following validly- and invalidly-cued memory tests in the 80% and 60% cue-reliability conditions. Circles and squares represent onset and offset latency, the light and dark colors represent validly- and invalidly-cued memory tests. Shading and error bars indicate ± 1 SEM. *, **, n.s represent significance level *p*<0.05, *p*<0.01, and not significant, respectively.

The online version of this article includes the following figure supplement(s) for figure 3:

**Figure supplement 1.** Time-frequency and saccadic activities after the memory test.

**Figure supplement 2.** Saccade biases associated with attentional re-orienting after the memory test as a function of behavioral performance.

**Figure supplement 3.** Differences between the gaze bias following expected and unexpected memory tests hold robust in subsampling analysis.

central test stimulus must reflect orienting in mind (i.e. to the memorized location of the tested item). We could evaluate this separately for memory tests that were certain (after 100% reliable cues), uncertain but conform the cue-induced expectation (after 80/60% *valid* cues), or unexpected (after 80/60% *invalid* cues).

As before, we considered spatial modulations in both alpha and gaze. For alpha lateralization, we found only weak spatial modulation in response to the test stimulus, with only the 80% cue-reliability condition showing a lateralization according to the location of the tested memory item that was larger following unexpected/invalid vs. expected/valid memory tests (*Figure 3a*, cluster *p*=0.048; see horizontal gray line in *Figure 3a*). For complementary time-frequency and topographical visualizations, see *Figure 3—figure supplement 1a*.

In contrast to EEG-alpha activity, when considering spatial modulations in gaze (*Figure 3b*), we found much clearer patterns after the memory test. The first thing that stands out is that we found a qualitatively different spatial modulation in our gaze marker of attentional orienting following memory tests when memory tests were fully anticipated (after 100% reliable cues) vs. uncertain (after 80/60% reliable cues). When the test was certain (100%), a significant saccade bias in the direction opposite from the memorized location of the tested item was observed (*Figure 3b*, cluster *p*=0.021, *p*=0.003, for imperative-100% and 100% conditions, respectively). This likely reflects 'return' saccades to the

center in response to the memory test stimulus from lingering gaze-position biases following the cue (i.e. a residual effect from the cue-induced modulation).

In contrast, and of key interest here, when the nature of the test stimulus was not 100% certain (in 80/60% cue-reliability conditions), we found a pronounced saccade bias towards the memorized location of the tested item (*Figure 3b*, 80%-valid block: cluster *p*=0.023, *p*<0.001; 60%-valid block: cluster *p*=0.009, *p*<0.001, for valid-cue and invalid-cue trials, respectively). Because the test stimulus was presented in the center, this saccade bias shows re-orienting to the mnemonic contents in mind, in response to the memory test. Critically, we found this re-orienting in mind not only following unexpected tests (for which the preceding retrocue initially directed attention to the other, non-tested, memory item during the delay), but also following expected tests.

As shown in the corresponding saccade-size-over-time plots in *Figure 3—figure supplement 1b*, consistent with what we observed following the cue, the difference between toward and away saccades following the test was again predominantly driven by saccades in the fixational micro-saccade range (<2°), and the time course (saccade bias) results hold even when exclusively considering fixational eye movements below 2 visual degrees (*Figure 2—figure supplement 1b*). Thus, just like mnemonic focusing after the cue, re-orienting after the memory test was also predominantly reflected in fixational micro-saccades, and not looking back at the original location of the memory items that were encoded at 6 degrees away from central fixation.

We also sorted patterns of gaze bias after the memory test by performance but could only establish a link between this gaze bias and RTs following expected memory tests in our 80% cue-reliability condition (cluster *p*=0.001, *Figure 3—figure supplement 2*). The lack of significant statistical differences in the remaining conditions may possibly reflect a lack of sensitivity (insufficient trial numbers) for this additional analysis.

Besides the striking difference in our gaze marker of internal orienting following memory tests that were certain (100% cue reliability) vs. uncertain (80/60% cue reliability), we also found revealing differences in response to memory tests that were expected (validly cued) vs. unexpected (invalidly cued), as we turn to and quantify in detail next.

## Unexpected memory tests prolong spatial re-orienting in working memory but do not delay the onset of re-orienting

As described above, and as shown in *Figure 3b*, we found prominent spatial biases in gaze in the direction of the memorized location of the tested memory item when the to-be-tested memory item was not certain until the test stimulus appeared (i.e. in the blocks with 80% and 60% reliable cues). We found this following both expected (validly cued) and unexpected (invalidly cued) memory tests. At the same time, we also observed a striking difference between these two types of trials. While our spatial marker of attention re-orienting in mind after the memory test started at a similar latency after expected and unexpected memory tests, the marker was prolonged following unexpected (invalidly cued) memory tests (*Figure 3b*).

We quantified this in two ways (as in *Wang and van Ede, 2024*): (a) comparing the time courses of our spatial gaze marker following expected and unexpected memory tests using a cluster-based permutation test (*Maris and Oostenveld, 2007*) and (b) using a jackknife latency analysis method that directly quantified the onset and offset of this re-orienting marker (as in *Smulders, 2010*, see also *Miller et al., 1998*). First, we found significant clusters when comparing the saccade bias following expected (validly cued) and unexpected (invalidly cued) memory tests, in both 80% and 60% cue-reliability blocks, as indicated by the black horizontal significance lines in *Figure 3b* (cluster *p*=0.003, *p*=0.025). Second, we directly compared the onset and offset latency of the saccade bias following expected and unexpected memory tests. For onset latency after the memory test, we found no statistical difference following expected and unexpected memory tests (80%: $t_{(21)}$ = –0.567, *p*=0.577, $BF_{01}$=3.877 in favor of the null hypothesis; 60%: $t_{(21)}$ = –0.440, *p*=0.665, $BF_{01}$=4.108 in favor of the null hypothesis). In contrast, we found a clear difference in the *offset* latency of this re-orienting bias in gaze (80%: $t_{(21)}$ = 2.200, *p*=0.039; 60%: $t_{(21)}$ = 3.409, *p*=0.003), consistent with a prolonged attentional re-orienting when the test did not match the cue-induced expectation of which memory item would likely be tested.

To rule out the possibility that the saccade-bias differences following expected and unexpected memory tests are caused by uneven trial numbers (200 vs. 50 trials in the 80% cue-reliability condition,

150 vs. 100 trials in the 60% cue-reliability condition), we ran a sub-sampling analysis where we sub-sampled the number of valid trials to match the number of invalid trials available per condition (averaging the results across 1000 random sub-samplings to increase reliability). As shown in *Figure 3—figure supplement 3*, this complementary sub-sampling analysis confirmed that our observed differences between the saccade bias following expected and unexpected memory tests in both 80% and 60% cue-reliability conditions are robust even when carefully matching the number of trials between validly cued (expected) and invalidly cued (unexpected) memory test.

For completeness, we also directly compared the valid-vs-invalid difference in offset latency between the 80% and 60% conditions. We found no evidence for a significant difference in the cue-validity effect on this prolonged re-orienting following memory tests in the 80% and 60% cue-reliability blocks ($t_{(21)}$ = 0.237, $p$=0.815, $BF_{01}$=5.962 in favor of the null hypothesis).

We additionally conducted fractional area latency (FAL) analysis on the comparison of the saccade bias following the memory test between valid- and invalid-cue trials to rule out the potential contribution of peak amplitude differences into the onset and offset latency differences (*Hansen and Hillyard, 1980*; *Kiesel et al., 2008*; *Luck, 2005*). Consistent with our jackknife-based latency analysis, the FAL analysis revealed a significantly prolonged saccade bias following the unexpected tests (the invalid-cue trials) vs. expected tests (the valid-cue trials) in both 80% and 60% cue-reliability conditions (411 ms vs. 463 ms, $t_{(14)}$ = 2.358, $p$=0.034; 417 ms vs. 468 ms, $t_{(15)}$ = 2.168, $p$=0.047; for 80% and 60% condition, respectively). Again, there was no significant statistical difference in onset latency following unexpected vs. expected tests. (346 ms vs. 374 ms, $t_{(14)}$ = 2.052, $p$=0.060; 353 ms vs. 401 ms, $t_{(15)}$ = 1.577, $p$=0.136; for 80% and 60% condition, respectively).

These results show not only how that the spatial saccade bias tracks attentional re-orienting in mind after the memory test, but also how the duration of such re-orienting is prolonged – but not delayed in onset – when having to return focus to the other, unexpected, memory item for report.

## Attention deployment after the memory test is not contingent on initial orienting after the cue

We have reported robust attentional deployment in working memory after the memory test, in trials where the cue was not 100% reliable. This could reflect that when the cue is not 100% reliable, participants may prioritize the cued item only partially, necessitating revisiting the cued item when it actually becomes tested (a second-stage 'verification' step that can be skipped with the cue is 100% reliable). At the same time, however, we also observed comparatively less orienting in response to the cue in these trials in which the cue is not 100% reliable. Accordingly, it remains possible that the re-orienting signal that we observed after the test is driven exclusively by trials where participants did not use the initial cue, but waited for the test before shifting attention to the relevant memory item for the first time (cf. *van Ede et al., 2019a*). In such a scenario, we should *only* observe attention deployment after the test stimulus in trials in which participants did *not* use the preceding retrocue.

While it is notoriously difficult to know for certain whether participants used the cue or not at the single-trial level, our saccade marker affords a handle on this. Unlike continuous alpha activity, saccades are events that can be classified on a single-trial level. Although alpha can also be analyzed as the single-trial level (e.g. *Macdonald et al., 2011*; *Wöstmann et al., 2019*; for a review, see *Kosciessa et al., 2020*), saccades offer the unique opportunity to split trials not by graded amplitude fluctuations but by discrete all-or-none events. We reasoned as follows: in the trial-average, we found a saccade bias between 200–600 ms after the cue (as in *Liu et al., 2022*) that was driven by more toward saccades. By zooming in on trials where we observed a toward saccade in this relevant window, we could isolate those trials that generated this bias in the trial average – designating them as the trials in which participants were *likely* to have used the cue. If initial orienting to the cue would eliminate the need to deploy attention after the test (at least for valid, expected memory tests), then our attentional deployment signal after the memory test should be attenuated (or even abolished) in these extracted trials.

As we show in *Figure 4a and b*, we found clear gaze signatures of attentional deployment in response to expected (valid) memory tests, no matter whether we had pre-selected trials in which we had also seen such deployment after the cue in gaze (cluster $p$s: 0.115, 0.041, 0.027,<0.001 for 80%-valid, 60%-valid, 80%-invalid, 60%-invalid trials, respectively), or not (cluster $p$s: 0.016, 0.009, 0.001,<0.001 for 80%-valid, 60%-valid, 80%-invalid, 60%-invalid trials, respectively). We only found

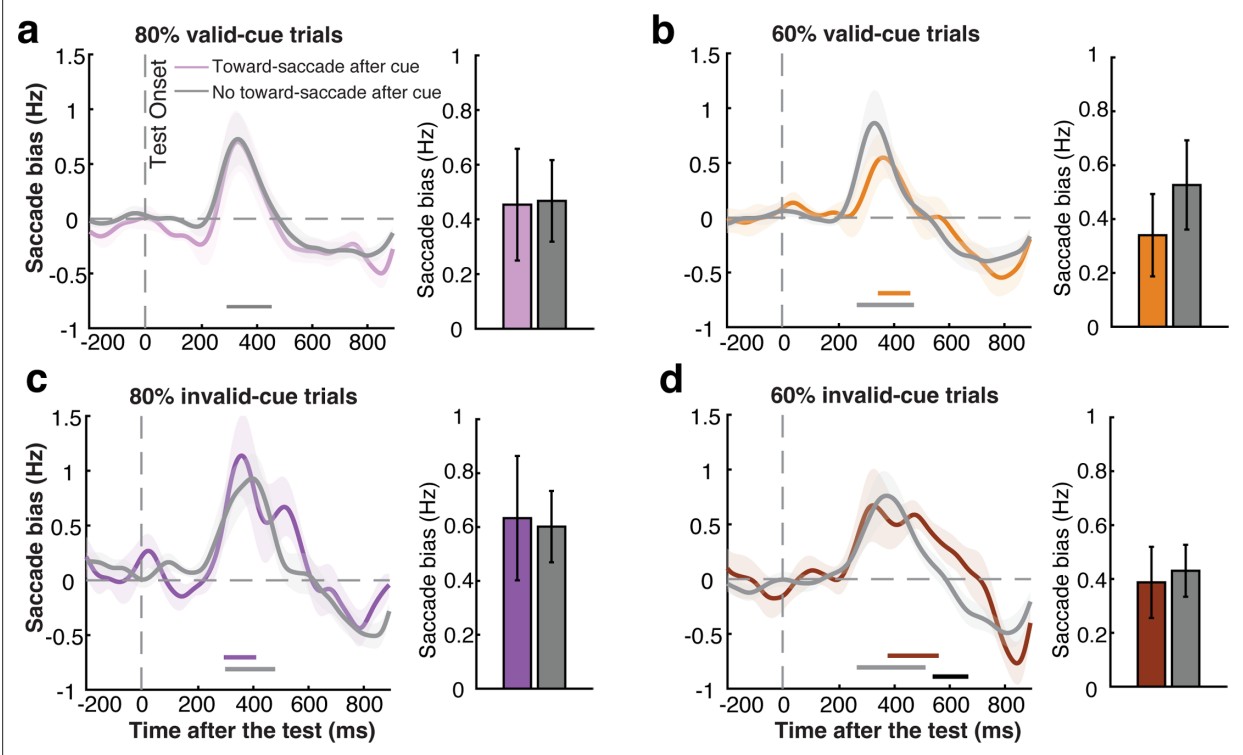

**Figure 4.** Re-orienting of attention after the memory test is not contingent on initial orienting after the cue. (**a**) Left panel: saccade bias towards the memorized location of the test item after memory test in trials with and without a toward saccade after the cue for validly cued trials in 80% cue-reliability block. Right panel: size of the test-locked saccade bias for the trials split by the preceding cue-locked saccades, averaged over the time window in which we observed the respective test-locked gaze bias in this condition. (**b**) As per panel a, but for 60% valid-cue trials. (**c**) and (**d**), as per panels **a** and **b**, but for invalidly cued trials. Colored horizontal lines above the x-axis indicate significant clusters (p<0.05). The black horizontal line above the x-axis in panel (**d**) indicates the significant difference cluster (p<0.05) between trials with and without a toward-saccade after the cue. Shading indicates ± 1 SEM.

one difference between the trials sorted by their response to the cue: with a more pronounced gaze bias to the memorized location of the tested item in the invalidly cued trials in the 60% cue-reliability condition (*Figure 4d*: cluster p=0.032). This may reflect the prolonged need to re-orient to the other memory item, when previously having relied more on the cue to prioritize the memory item that did not end up being tested in these invalidly cued trials.

To increase sensitivity for assessing whether the gaze bias following the memory test was different in those trials in which participants already showed a toward saccade following the retrocue, we zoomed in on the time windows in which we observed the respective test-locked gaze biases in each condition. Again, we found no evidence that the test-locked bias was contingent on the prior use of the cue (bar graphs *Figure 4a–d*: ts=−0.226,−0.047, −1.717,−0.399, ps=0.824, 0.963, 0.101, 0.694, $BF_{01}$=3.930, 5.856, 1.613, 5.673 in favor of the null hypothesis across all four conditions).

These results suggest that observers deploy attention to the memorized location in response to the memory test, whether or not they had already oriented their attention after the cue. In other words, this suggests that deployment of attention after the memory test may reflect an additional verification stage following expected memory tests (in valid-cue trials), and a genuine re-orienting of attention to the other memory item following unexpected memory tests (in invalid-cue trials).

## Discussion

Our data reveal three central sets of results that we summarize in *Figure 5*. First, building on ample prior studies that have tracked attentional orienting following retrocues, we show how two spatial markers of internal attentional orienting – alpha lateralization and saccade-direction bias – each tracks attentional orienting in working memory in a graded fashion, according to cue reliability. Complementing

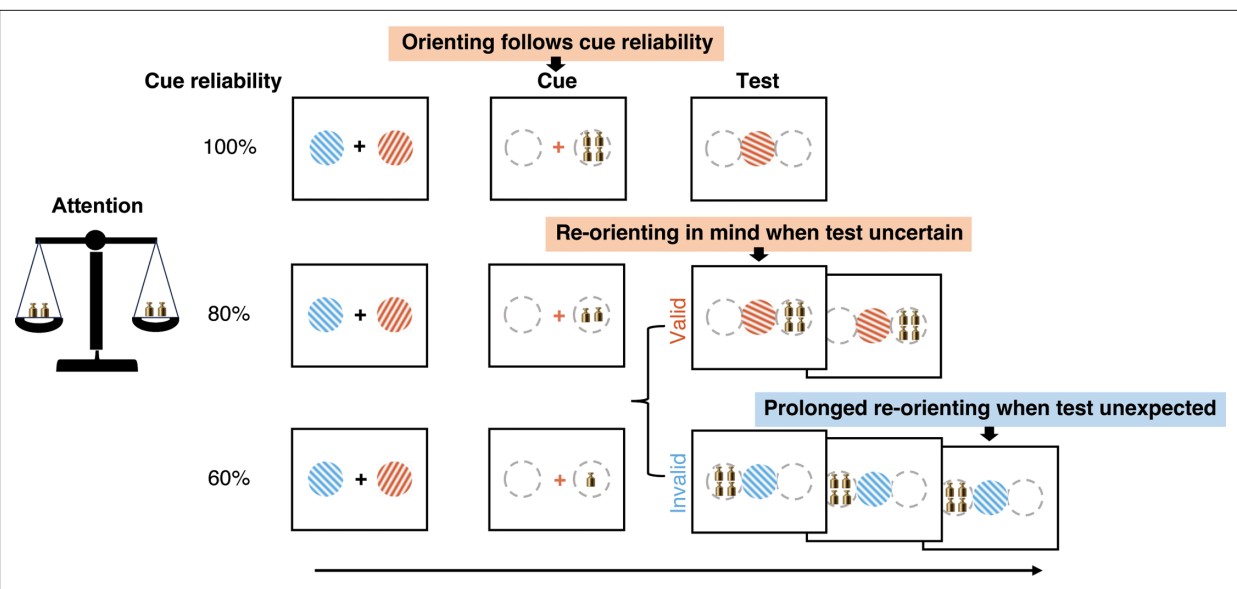

**Figure 5.** Schematic summary of the main findings. Schematic summary of our central findings regarding dynamic attentional orienting and re-orienting after the cue and the memory test, as a function of cue reliability and test expectations.

this more established line of research, we here unveil similar attentional deployment within the spatial layout of working memory in response to the memory test, but only when there remained uncertainty about which memory item would be tested. Finally, when the memory test probed the unexpected memory item (in invalidly cued trials), we found that attentional re-orienting did not take off later but did become prolonged, consistent with the slower memory-guided behavior after unexpected memory tests. We discuss each finding in turn.

Consistent with ample prior studies (*Astle et al., 2012*; *Griffin and Nobre, 2003*; *Lepsien et al., 2005*; *Matsukura et al., 2007*; *Rerko and Oberauer, 2013*; *van Moorselaar et al., 2015*) (for reviews, see *Souza and Oberauer, 2016*; *van Ede and Nobre, 2023*), our behavioral data showed clear cueing benefits on memory-guided behavior, as most clearly demonstrated by better and faster performance following memory tests that were preceded by a valid vs. invalid retrocue. Moreover, at least for RT, we also found a larger cue-validity effect when cues were more reliable, suggesting a graded allocation of internal selective attention that depended on cue reliability (cf. *Berryhill et al., 2012*; *Ester and Pytel, 2023*; *Fu et al., 2022*; *Gözenman et al., 2014*; *Günseli et al., 2019*; *Gunseli et al., 2015*; *Shimi et al., 2014*). More direct evidence for this graded attentional allocation came from our two continuous markers of internal attentional orienting. First, we found a robust spatial lateralization of alpha activity in all conditions, but also found that this modulation was more pronounced following more reliable cues. This is consistent with several prior studies demonstrating graded alpha lateralization according to cue reliability (*Fu et al., 2022*; *Gould et al., 2011*; *Günseli et al., 2019*; *Haegens et al., 2012*). In addition to this more established marker, we here also tracked internal attentional allocation via spatial biases in saccade directions, building on several prior studies from our lab that have demonstrated spatial biases in microsaccade directions as a reliable marker of internal orienting of attention (*de Vries et al., 2023*; *Draschkow et al., 2022*; *Liu et al., 2022*; *van Ede et al., 2021*; *van Ede et al., 2020*; *van Ede et al., 2019a*; *Wang and van Ede, 2024*). To our knowledge, this is the first demonstration of a graded modulation of this gaze marker of attention shifts within the spatial layout of visual working memory.

While we observe clearly graded modulations in the trial average, it is notoriously hard to know whether attentional allocation was also graded at the single-trial level (*Stokes et al., 2015*). For example, following 80% reliable cues, it could be that participants consistently deployed 80% of their full 'attention potential', or deployed their full attention potential in 80% of the trials. Disambiguating these alternatives is challenging, and beyond the scope of our work. In addition, while it is of interest to understand to what extend our two markers track the same vs. distinct components of internal

attention, this too is beyond the scope of our current study and instead, we refer the reader to *Liu et al., 2022*.

A central novelty of our study is the demonstration of attentional re-orienting after the working-memory test – and its dependence on the reliability of the preceding retrocue. While ample studies from the study of external attention have considered spatial re-orienting of attention to unexpected perceptual targets (*Carrasco, 2018*; *Corbetta et al., 2002*; *Corbetta et al., 2000*; *Doricchi et al., 2010*; *Posner, 1980*; *Posner et al., 1982*; *Posner and Cohen, 1984*), the study of spatial attentional deployment of working memory has predominantly focused on spatial orienting in response to retro-cues, without additionally considering *re*-orienting in response to expected/unexpected memory tests (the analog of expected/unexpected targets in perceptual-attention studies).

To date, several prior studies have demonstrated how attention can dynamically orient and re-orient to previously encoded memory items in anticipation of the memory test, such as following sequential retrocues (*de Vries et al., 2020*; *de Vries et al., 2018*; *Rose et al., 2016*; *Yu et al., 2020*), or following expectations that designated one item relevant early and another late (*van Ede et al., 2017*; *Zokaei et al., 2019*). In these studies, orienting/re-orienting happened while anticipating which memory item would become tested. In contrast, our findings of attentional re-orienting specifically concern spatial re-orienting in response to the memory test; not in anticipation of this test. While ample prior studies have considered processes related to retrieval at the memory test (*Ester and Pytel, 2023*; *Nobre et al., 2007*; *Schneider et al., 2015*; *Souza et al., 2016*), or reselection at memory encoding (*Chen et al., 2019*; *Chen and Wyble, 2015*; *Fu et al., 2023*), we uniquely focused on deployment of spatial attention to the tested memory item (not general, non-spatial cognitive processes related to retrieval, or attentional reselection to decide what features to be encoded into working memory). We were in a unique position to isolate such spatial deployment of attention following the test, because our memory test was always centrally presented while our memory items were presented lateralized.

Regarding our findings in response to the memory test, we have three key findings that we discuss in turn. First, we show that attentional deployment after the memory test only occurs when the cue leaves some uncertainty regarding which memory item will be tested. When the cue is 100% reliable, participants may drop the other memory item (*Astle et al., 2012*; *Gözenman et al., 2014*; *Gressmann and Janczyk, 2016*; *Griffin and Nobre, 2003*; *Gunseli et al., 2015*; *Oberauer, 2001*; *Pertzov et al., 2013*; *van Moorselaar et al., 2015*; *Williams et al., 2013*) or transform the relevant item into a different format (cf. *Larocque et al., 2014*; *LaRocque et al., 2013*; *Lewis-Peacock et al., 2012*; *Myers et al., 2017*; *Rose et al., 2016*; *Stokes, 2015*; *Wolff et al., 2017*) that does not invoke spatial revisiting after the memory test. For example, participants may transform the relevant item into the appropriate action plan, when 100% certain (*Henderson et al., 2022*; *Nasrawi et al., 2023*; *Nasrawi and van Ede, 2022*; *Olivers and Roelfsema, 2020*; *Schneider et al., 2020*; *van Ede et al., 2019b*). In contrast, when the other item cannot be dropped, in the case of cues that leave some uncertainty, the memory items likely preserve their spatial organization, inviting spatial deployment of attention following the memory test.

Second, we found attentional deployment in mind not only following unexpected memory tests – for which attention required to be *re*-directed to the unexpected memory item – but also following expected memory tests. This suggests that, when both items could become tested, participants continued to rely on the memory test to revisit the appropriate memory item, even when they had already attended it after the cue. In other words, participants may engage in a second-stage verification ('double checking') in mind, whenever cues do not predict memory tests with certainty. Indeed, when sorting the trials by whether participants were likely to have already attended the cued memory item following the preceding retrocue, we found no evidence that having previously attended the memory item diminished the attentional deployment after the test.

Third, we found a clear difference in spatial deployment of attention following expected and unexpected memory tests. Specifically, spatial deployment (re-orienting) of attention in mind was prolonged when the test prompted a report for the memory item that was not expected to become tested. This is consistent with the slowing of memory-guided behavior that is typically found after unexpected memory tests (*Astle et al., 2012*; *Griffin and Nobre, 2003*; *Lepsien et al., 2005*; *Matsukura et al., 2007*; *Rerko and Oberauer, 2013*; *van Moorselaar et al., 2014*, for reviews, see *Souza and Oberauer, 2016*; *van Ede and Nobre, 2023*), as also observed in the current study. At the same time, we found no evidence that unexpected memory tests resulted in delayed attentional allocation to the

appropriate item, as would be expected if the internal focus of attention first had to be 'disengaged' before it could be relocated. Thus, while unexpected memory tests do not delay the process of re-orienting in mind to the appropriate memory content, attentional deployment did become prolonged when re-focusing the unexpected memory item, likely reflecting prolonged effort to extract the relevant information from the memory item that was not expected to be tested. However, there may also be alternative accounts for this observation, such as suppressing a learned tendency to saccade in the direction of the expected item following an unexpected memory test.

By considering two markers of spatial orienting/re-orienting of internal selective attention, we were in the unique position to assess their utility for tracking attentional focusing in memory. Though we found both EEG-alpha and gaze markers to robustly track spatial orienting following the retrocue, only gaze showed a clear pattern of re-orienting in response to the test stimulus. In our task, the test stimulus was a large central stimulus that may have attenuated alpha activity, possibly making it challenging to find spatial modulations in alpha activity (see e.g. *van Ede et al., 2014*), without similarly affecting gaze. In future studies, it will prove interesting to more carefully delineate such factors and to more carefully compare the utility and function of both markers.

Collectively, our findings delineate the dynamics of spatial orienting and re-orienting of attention in visual working memory following attentional retrocues and memory tests. By testing working memory items centrally, our data uniquely reveal a second stage of internal attentional deployment following expected and unexpected memory tests. We unveil that participants revisit tested memory content even when contents were already previously attended – suggesting a second verification ('double checking') in mind. Moreover, when tests prompted unexpected memory content, we found that re-orienting in mind became prolonged, putatively reflecting a prolonged effort to re-sample unexpected content from memory. These findings bring attentional re-orienting to the domain of working memory and underscore the relevance of studying attentional dynamics in memory not only following explicit attention cues, but also following ensuing memory tests that can invoke a revision or verification of our internal focus.

## Methods

### Participants

Twenty-four healthy human volunteers participated in the experiment (age ranges from 19 to 28; 17 females and 7 males, all right-handed). Sample size was set as a multiple of 6 for counterbalancing, and was set a-priori to n=24 based on previous publications from our lab that had similar designs and focused on similar EEG and eye-tracking outcome variables (e.g. *de Vries et al., 2023*; *Draschkow et al., 2022*; *Liu et al., 2022*; *van Ede et al., 2021*; *van Ede et al., 2020*; *van Ede et al., 2019a*). Two participants were excluded from the eye-tracking dataset due to the poor quality of their eye-tracking data caused by wearing glasses; one participant was excluded from the EEG dataset due to large number of artifacts in their EEG signal. The experimental procedures were reviewed and approved by the Scientific and Ethical Review Board (VCWE) of the Faculty of Behavior & Movement Sciences, VU University Amsterdam (reference number: VCWE-2021–009). All participants provided written informed consent, and consent to publish prior to participation and were reimbursed 10 euros per hour as compensation for their time.

### Task and procedure

Participants performed an internal selective-attention task (*Figure 1a*) whereby a retrospective cue (i.e. retrocue) informed participants which of the two previously encoded memory items was either certain (100%) or more likely (80/60%) to become tested for an ensuing behavioral orientation-reproduction report.

Each trial (*Figure 1a*) began with an encoding display containing two to-be-memorized gratings (with different colors and orientations) for 200 ms, followed by an initial delay period with a central fixation cross that remained on the screen for 800 ms. Then, the central fixation cross changed its color to match the color of one of the memorized gratings for 200 ms, acting as the attentional cue that predicted which memory item would most likely become tested after another delay. After the second delay interval (1300 ms) following the cue, participants were presented with the centrally presented memory test with a random orientation and were instructed to reproduce the precise orientation of

the tested memory item by moving the mouse with their dominant hand (right hand) and terminating the report by clicking the mouse button. A feedback score ranging from 0 to 100 was given after each report (with score of 100 reflecting the minimum error of 0, and score of 0 reflecting the maximum error of 90 degrees).

In a block-wise manipulation, we employed four cue-reliability conditions: informative 100%, 80%, and 60% cue-reliability conditions, and an imperative 100% cue-reliability condition. In the informative cue conditions, we always indicated which memory item to report via the color of the memory test. In the 100% condition, the color of the cue predicted the color of the upcoming test stimulus with 100% reliability. In the 80% condition, the cue predicted the color of the upcoming test correctly in 80% of trials (valid-cue trials), while in the remaining 20% of trials the cue was invalid and the uncued memory item was tested instead (invalid-cue trials). The 60% condition was identical except now the cue was only 60% reliable (60% valid-cue trials, 40% invalid-cue trials). Because in all the three informative-cue conditions, participants needed to make a report based on the color of the test grating, it was possible that participants could neglect the attentional cue even when it was 100% reliable. We, therefore, added a fourth condition, the imperative 100% cue-reliability condition. In this block, the cue was again 100% reliable, but the color of the test grating was always black such that participants were forced to use the color of the cue to know which memory item they would need to report.

Each participant completed two sessions, one informative-cue session and one imperative-cue session. We did this, so that participants would only have to switch once between blocks with a colored vs. a black memory test. Before starting each session, participants practiced 20 trials of the 100%-reliable cue with the respective testing style (color test for informative session, black test for imperative session) for about 5 minutes. The session sequence was counterbalanced across participants. For the informative-cue session, participants finished three blocks of 250 trials each for the 100%, 80%, and 60% conditions, respectively, and the sequence of different cue-reliability was pseudorandomized and counterbalanced across participants. For the imperative-cue session, participants finished one block of 250 trials for the 100%-reliable cue condition. In total, each participant completed 1000 trials lasting about 2 hours in total.

## Apparatus and stimuli

Stimuli were presented using MATLAB (R2020a; MathWorks) and the Psychophysics Toolbox (version 3.0.16, *Brainard, 1997*) on a LED monitor. The stimuli were presented on a 23-inch (58.42 cm) screen running at 240 Hz and a resolution of 1920 by 1080. Participants were seated 70 cm from the screen with their head resting on a chin rest to ensure stable eye-tracking.

A gray background (RGB = 128, 128, 128) was maintained throughout the experiment, along with a fixation cross (0.7°) presented in the center of the screen. Memory items and test items were sine-wave gratings presented at 100% contrast, with a diameter of 4.6°, spatial frequency of 0.032 cycles per pixel, and a phase of 90°. The memory items were presented at 6° eccentricity to the left and right of central fixation, and for each trial, the orientations were randomly selected from a uniform distribution of orientations (from 1 to 180°). The color of one of the memory gratings was blue (RGB: 21, 165, 234) and the other one was orange (RGB: 234, 74, 21), and color-location mapping was varied across trials. The memory test item (at the report stage) was presented in the center of the screen, with the same size as the memory gratings, and its initial orientation was randomly selected. The color of the memory test item was either blue or orange in the informative-cue blocks (100%, 80%, and 60%), and was always black in the imperative-cue block (100%). Gratings in the encoding display were equally likely to be cued for report, and the left/right location of the cued memory item was randomized across trials.

## Analysis of behavioral data

Reproduction errors were defined as the absolute difference (in degrees) between the reported orientation and the actual orientation of the tested memory item. RT was defined as the time from the memory test onset to the report completion. Trials with RTs longer than 4 seconds were excluded from further analysis.

## EEG acquisition and pre-processing

EEG signals were recorded using a 64-channel Biosemi system (1024 Hz sampling rate), with active electrodes distributed across the scalp using the international 10–20 positioning system. The CMS and DRL, embedded in the cap on the left and right side of the POz, were used as the online reference. We re-referenced the signal offline to the average of both mastoids. To monitor for eye movement- and blink artifacts, we placed two external electrodes horizontally next to the left and right eye, and two electrodes above and below the right eye. These EOG measurements were only included for data cleaning using ICA (described below). We additionally included eye-tracking using an Eyelink that we used to extract our eye-tracking outcome signal of interest (as we turn to later).

Offline data analyses were conducted in MATLAB through a combination of Fieldtrip (*Oostenveld et al., 2011*) and custom code. After re-referencing, the continuous EEG signal was epoched from −200 to +1500 ms relative to the retrocue onset for the cue-locked analyses (or from −200 to 900 ms relative to memory test onset for the test-locked EEG analyses). A fast independent component analysis (ICA), as implemented in the Fieldtrip, was then applied to the EEG epochs to remove components associated with blinks and eye movements. ICA components that captured eye blinks and horizontal eye movements were removed after inspecting the correlations between each component with the VEOG (vertical electrooculography: the difference signal between the electrodes above and below the right eye) and the HEOG (horizontal electrooculography: the difference signal between the electrodes horizontally next to the left and right eye) signal, respectively. A visual inspection method by using 'ft_rejectvisual.m' function in Fieldtrip with the 'summary' method was then applied and trials with exceptionally high variance were removed as artifact trials. After trial removal, we had on average 241 (sd = 12.0), 244 (sd = 6.8), 243 (sd = 8.7), 243 (sd = 7.2) trials left for informative 100%, 80%, 60%, and imperative 100% conditions, respectively. Trial removal was done once on the full datasets without knowledge of the conditions to which individual trials belonged.

## EEG time-frequency analysis

For EEG data time-frequency decomposition, we first performed a surface Laplacian transform to increase the spatial resolution of the EEG. We then decomposed the clean EEG signal epochs into a time-frequency representation using a short-time Fourier transform of Hanning-tapered data as implemented in Fieldtrip, using the 'ft_freqanalysis.m' function. A 300 ms sliding time window was used to estimate the spectral power between 2 and 40 Hz (in steps of 1 Hz) that was advanced over the data in steps of 10 ms.

For the calculation of alpha lateralization, we first extracted the spectral power in the alpha band (8–12 Hz). We then calculated alpha lateralization as the normalized percentage change between trials in which the memorized location of the cued memory item was contralateral vs. ipsilateral to the posterior electrodes of interest: ((contralateral - ipsilateral)/(contralateral +ipsilateral))×100 (as also done in *Liu et al., 2022*; *van Ede et al., 2019a*). We then averaged these contrasts across the left and right electrode clusters of interest. Consistent with previous studies (*Adam et al., 2018*; *Liu et al., 2022*; *Wang et al., 2020*; *Wang et al., 2018*), the calculation of the alpha lateralization focused on posterior electrode clusters (left: O1, PO7, PO3, P9, P7, P5, P3, P1, O2; right: PO8, PO4, P10, P8, P6, P4, P2). In addition, to obtain topographical maps of lateralization, we contrasted alpha power for left and right attention trials (i.e. ((left-right)/(left +right))×100) for all electrodes.

## Eye-tracking acquisition and pre-processing

The eye tracker (EyeLink 1000, SR Research SR) was positioned ~5 cm in front of the monitor on a table(~65 cm away from the eyes). Horizontal and vertical gaze position was continuously recorded for single eye at a sampling rate of 1000 Hz. Before recording, the eye tracker was calibrated through the built-in calibration and validation protocols from the Eye-Link software. Participants were asked to keep their chin on the chin rest for the entire block after calibration.

Offline, eye-tracking data were converted from the .edf to the .asc format, and were read into MATLAB through Fieldtrip. Blinks were marked by detecting clusters of zeros in the eye-tracking data. Then the data from 100 ms before to 100 ms after the detected blink clusters were turned to Not-a-Number (NaN) and excluded from further analysis to eliminate residual blink artifacts. After blink rejection, data were epoched, once relative to the cue onset and once relative to the memory test onset (equivalent to our EEG analyses).

## Saccade detection

We employed a velocity-based method to detect saccades. This method was built on other established velocity-based methods for saccade detection (e.g. *Engbert and Kliegl, 2003*), and has been successfully applied for saccade detection in working memory retrocue tasks similar to the current study (*de Vries et al., 2023*; *Liu et al., 2023*; *Liu et al., 2022*). Since the memory items in the current experiment were always presented horizontally (i.e. center-left, and center-right), our saccade detection focused on the horizontal channel of the eye data – an approach we have previously demonstrated to be effective in capturing directional biases in saccades (see *Liu et al., 2022*).

We first computed gaze velocity by measuring the distance between consecutive temporal gaze positions. To enhance precision and diminish noise, we applied temporal smoothing to the velocity using a Gaussian-weighted moving average filter with a 7 ms sliding window, utilizing the 'smooth-data' function in MATLAB. We then determined the onset of a saccade as the first sample where the velocity surpassed a trial-based threshold set at five times the median velocity. To prevent multiple classifications of the same saccade, we enforced a minimum delay of 100 ms between successive saccades. Saccade magnitude and direction were determined by assessing the disparity between pre-saccade gaze position (from −50–0 ms before threshold crossing) and post-saccade gaze position (from 50 to 100 ms after threshold crossing). Finally, each identified saccade was categorized as 'toward' or 'away' based on its direction (left/right) and the side of the cued memory item (left/right). 'Toward' denoted a saccade aligned with the memorized location of the cued item, while 'away' indicated a saccade opposite to the memorized location of the cued item. Following saccade identification and labeling, we quantified the time courses of saccade rates (in Hz) using a sliding time window of 50 ms, progressing in 1 ms increments. For our main analyses, we focused directly on the spatial bias in saccades, calculated simply as the differences (in Hz) between toward and away saccades. We also separately show toward and away saccade rates.

## Saccade latency analysis

To compare the latency of the time-series data of the spatial saccade bias after the memory test, we applied a simplified jackknife method (as outlined in *Smulders, 2010*) to compare the latency of onset and offset of the saccade bias between validly and invalidly cued memory tests in the reporting stage. The onset latency and offset latency were defined by the time point that the amplitude reaches or drops to 50% of the peak value, respectively.

In addition to the jackknife-based latency analysis, we further applied a fractional area latency (FAL) method to the saccade bias comparison between validly and invalidly cued memory tests to rule out the contribution of the peak amplitude difference into the onset and offset latency difference (*Hansen and Hillyard, 1980*; *Kiesel et al., 2008*; *Luck, 2005*). We first defined the onset and offset latency of the saccade bias as the first time point at which 25% or 75% of the total area of the component has been reached, relative to a lower boundary of a difference of 0.3 Hz between toward and away saccades (to remove the influence of noise fluctuations in our difference time course below this lower boundary). The extracted onset and offset latency for all participants was then compared using paired-samples t-tests.

## Saccade-toward trials extraction

To assess whether our findings of re-orienting at the test phase were contingent on the initial orienting following the cue, we aimed to separate trials based on saccades following the cue. To extract those trials in which we could have increased confidence that participants had used the cue at the single-trial level, we classified trials into whether a saccade in the direction toward the cued memory item was detected or not in the critical time window of 200–600 ms after the cue (as in *Liu et al., 2022*). Specifically, after getting the usable eye traces for each epoch, we extracted trials with discernible saccades (gaze shifts) detected in the 200–600 ms post-cue period and in which the direction of the first saccade matched the memorized location of the cued item as the 'toward-saccade after the cue' trials (80% cue-reliability blocks: 65.5±27.8 trials; 60% cue reliability blocks: 60.8±27.8 trials). Alternatively, all other remaining trials were labeled as the 'no toward-saccade after the cue' rials (80%: 184.5±27.8 trials; 60%: 189.2±27.8 trials). Following this trial separation rule based on the data at the cue stage, we then compared saccade biases in response to the test stimulus.

## Statistical analysis

Statistical evaluation for the behavioral data was achieved first by one-way repeated-measures ANOVAs on the absolute reproduction error and RTs of all cue-reliabilities (imperative 100%, informative 100%, 80%, and 60% blocks), respectively. For the comparison between trials with valid vs. invalid cues (which were only available in the 80% and 60% blocks), paired-sample t-tests between validly and invalidly cued trials were applied to the reproduction error and the response time, respectively. To test whether cue-validity effects depended on cue reliability, we additionally employed as two-way repeated measures ANOVA with the factors of cue-validity (valid/invalid) and cue-reliability (80/60%). A Bonferroni method was applied for multiple-comparison correction. All reported *p* values are Bonferroni corrected.

To statistically assess the temporal profiles of the spatial modulations in EEG activity (alpha lateralization) and eye movements (spatial saccade bias), we employed a cluster-based permutation approach (*Maris and Oostenveld, 2007*) using the fieldtrip 'ft_timelockstatistics' function with 'montecarlo' method. This method is well-suited for evaluating the consistency of data patterns at multiple adjacent data points under a single permutation distribution of the largest cluster, effectively bypassing the multiple-comparisons problem. We conducted 10,000 permutations to construct the permutation distribution of the largest cluster that would be obtained by chance and identified clusters using Fieldtrip's default settings, which involve grouping temporally adjacent data points with the same sign that were statistically significant in a mass univariate t-test at a two-sided alpha level of 0.05. Cluster size was defined as the sum of all *t* values within a cluster. We applied this procedure to test the time-series of spatial biases (alpha lateralization and saccade bias after cue) against 0 and to directly compare spatial biases between validly and invalidly cued memory tests (saccade bias after test).

In addition to the statistical evaluations that considered the full time-courses, we also used a-priori knowledge from prior studies to zoom in on relevant time windows for both outcome measures. This served mainly to ease direct comparison between our cue-reliability conditions. For alpha lateralization, we zoomed in on the a-priori-defined 400–800 ms post-cue window, based on the same window used for visualization in *Liu et al., 2022*; *van Ede et al., 2019a*. Likewise, for the spatial saccade bias, we zoomed in on the a-priori-defined 200–600 ms post-cue window based on a prior study from our lab that revealed how this is the critical window after cue onset in which saccade directions are biased toward the memorized location of the cued memory target (see *Liu et al., 2022*).

For the saccade bias latency comparison between validly and invalidly cued memory tests, the estimates of latency from the employed jackknife method (*Smulders, 2010*) were compared between validly and invalidly cued memory tests using paired-samples t-tests.

Where relevant, we also employed Bayesian analysis (*Rouder et al., 2017*) to evaluate how many times the null hypothesis was favored compared to the hypothesis that a difference between the two conditions might exist.

## Acknowledgements

This work was supported by a NWO Vidi Grant from the Dutch Research Council (14721) and an ERC Starting Grant from the European Research Council (MEMTICIPATION, 850636) to FvE.

## Additional information

### Funding

| Funder | Grant reference number | Author |
|---|---|---|
| NWO Vidi Grant | 14721 | Freek van Ede |
| ERC Starting Grant | MEMTICIPATION850636 | Freek van Ede |

The funders had no role in study design, data collection and interpretation, or the decision to submit the work for publication.

## Author contributions
Sisi Wang, Conceptualization, Data curation, Formal analysis, Validation, Investigation, Visualization, Methodology, Writing – original draft, Writing – review and editing; Freek van Ede, Conceptualization, Resources, Data curation, Formal analysis, Supervision, Funding acquisition, Validation, Investigation, Visualization, Methodology, Project administration, Writing – review and editing

## Author ORCIDs
Sisi Wang (iD) https://orcid.org/0000-0002-9730-438X
Freek van Ede (iD) http://orcid.org/0000-0002-7434-1751

## Ethics
The experimental procedures were reviewed and approved by the Scientific and Ethical Review Board (VCWE) of the Faculty of Behavior & Movement Sciences, VU University Amsterdam (reference number: VCWE-2021-009). All participants provided written informed consent, and consent to publish prior to participation and were reimbursed 10 euros per hour as compensation for their time.

Reviewer #1 (Public review): https://doi.org/10.7554/eLife.100532.3.sa1
Reviewer #2 (Public review): https://doi.org/10.7554/eLife.100532.3.sa2
Reviewer #3 (Public review): https://doi.org/10.7554/eLife.100532.3.sa3
Author response https://doi.org/10.7554/eLife.100532.3.sa4

## Additional files

### Supplementary files
MDAR checklist

### Data availability
All data and analysis scripts are publicly available in the OSF: https://osf.io/5wexp/.

The following dataset was generated:

| Author(s) | Year | Dataset title | Dataset URL | Database and Identifier |
| --- | --- | --- | --- | --- |
| Wang S, de van EF | 2024 | Cue_Validity | https://osf.io/5wexp/ | Open Science Framework, 5wexp |

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
