## [Editor Report · eLife Assessment]

This **important** study provides significant insights into the dynamics of attentional re-orienting within visual working memory, demonstrating how expected and unexpected memory tests influence attention focus and re-focus. The evidence supporting these conclusions is **convincing**, with the use of state-of-the-art methodologies. This work will be of interest to cognitive neuroscientists studying attention and memory.

---

## [Referee Report · Reviewer #1 (Public review)]

This study provides significant insights into the dynamics of attentional re-orienting within visual working memory, demonstrating how expected and unexpected memory tests influence attention focus and re-focus. The evidence supporting these conclusions is convincing, with the use of appropriate and validated methodologies, including behavioral measures, EEG, and eye tracking, that are in line with current state-of-the-art practices. This work will be of particular interest to cognitive neuroscientists studying attention and memory processes.

Thank you for the detailed revisions. I am pleased to see that the manuscript now effectively addresses every point I raised. The clarification between microsaccades and saccades greatly enhances transparency regarding the eye movement data. The inclusion of time-frequency plots and topographic maps for the working-memory test phase further improves the comprehensiveness of the alpha lateralization results, despite the relative lack of alpha effects at that stage. Moreover, the implementation of the Fractional Area Latency analysis successfully rules out amplitude-related confounds in the saccade bias latency measurements. Finally, the clear reporting of the statistical analyses for significant saccade bias further strengthens the reliability of the findings.

Overall, I appreciate the thorough and thoughtful response, and I believe that all my concerns have been successfully addressed.

---

## [Referee Report · Reviewer #2 (Public review)]

Summary:

This study utilized EEG-alpha activity and saccade bias to quantify the spatial allocation of attention during a working memory task. The findings indicate a second stage of internal attentional deployment following the appearance of memory test, revealing distinct patterns between expected and unexpected test trials. The spatial bias observed during expected test suggests a memory verification process, whereas the prolonged spatial bias during unexpected test suggests a re-orienting response to the memory test. This work offers novel insights into the dynamics of attentional deployment, particularly in terms of orienting and re-orienting following both the cue and memory test.

Strengths:

The inclusion of both EEG-alpha activity and saccade bias yields consistent results in quantifying the attentional orienting and re-orienting processes. The data clearly delineate the dynamics of spatial attentional shifts in working memory. The findings of a second stage of attentional re-orienting may enhance our understanding of how memorized information is retrieved.

Weaknesses:

The authors addressed the identified weaknesses in a thorough revision during the review process.

---

## [Referee Report · Reviewer #3 (Public review)]

Summary:

Wang and van Ede investigate whether and how attention re-orients within visual working memory following expected and unexpected centrally presented memory tests. Using a combination of spatial modulations in neural activity (EEG-alpha lateralization) and gaze bias quantified as time courses of microsaccade rate, the authors examined how retro cues with varying levels of reliability influence attentional deployment and subsequent memory performance. The conclusion is that attentional re-orienting occurs within visual working memory, even when tested centrally, with distinct patterns following expected and unexpected tests. The findings provide new value for the field and are likely of broad interest and impact, by highlighting working memory as an action-bound process (in)dependent on (an ambiguous) past.

Strengths:

The study uniquely integrates behavioral data (accuracy and reaction time), EEG-alpha activity, and gaze tracking to provide a comprehensive analysis of attentional re-orienting within visual working memory. As typical for this research group, the validity of the findings follows from the task design that effectively manipulates the reliability of retro cues and isolates attentional processes related to memory tests. The use of well-established markers for spatial attention (i.e. alpha lateralization) and more recently entangled dependent variable (gaze bias) is commendable. Utilizing these dependent metrics, the concise report presents a thorough analysis of the scaling effects of cue reliability on attentional deployment, both at the behavioral and neural levels. The clear demonstration of prolonged attentional deployment following unexpected memory tests is particularly noteworthy, although there are no significant time clusters per definition as time isn't a factor in a statistical sense, the jackknife approach is convincing. Overall, the evidence is compelling, allowing the conclusion of a second stage of internal attentional deployment following both expected and unexpected memory tests, highlighting the importance of memory verification and re-orienting processes.

Weaknesses:

I want to stress upfront that these are not specific to the presented work and do not affect my recommendation to offer the report to the public in its present form.

The sample size is consistent with previous studies, a larger sample could enhance the generalizability and robustness of the findings. The authors acknowledge high noise levels in EEG-alpha activity, which may affect the reliability of this marker. This is a general issue in non-invasive electrophysiology that cannot be handled by the authors but an interested reader should be aware of it. Effectively, the sensitivity of the gaze analysis appears "better" in part due to the better SNR. The latter also sets the boundaries for single trial analyses as the authors correctly mention. In terms of generalizability, I am convinced that the main outcome will likely generalize to different samples and stimulus types. Yet, as typical for the field, future research could explore different contexts and task demands to validate and extend the findings. The authors provide here how and why (including sharing of data and code).

Comments on revisions:

Really nice work, Thank you!

---

## [Author Response]

The following is the authors’ response to the original reviews.

**Public Reviews:**

**Reviewer #1 (Public Review):**
In the study "Re-focusing visual working memory during expected and unexpected memory tests" by Sisi Wang and Freek van Ede, the authors investigate the dynamics of attentional re-orienting within visual working memory (VWM). Utilizing a robust combination of behavioral measures, electroencephalography (EEG), and eye tracking, the research presents a compelling exploration of how attention is redirected within VWM under varying conditions. The research question addresses a significant gap in our understanding of cognitive processes, particularly how expected and unexpected memory tests influence the focus and re-focus of attention. The experimental design is meticulously crafted, enabling a thorough investigation of these dynamics. The figures presented are clear and effectively illustrate the findings, while the writing is concise and accessible, making the complex concepts understandable. Overall, this study provides valuable insights into the mechanisms of visual working memory and attentional re-orienting, contributing meaningfully to the field of cognitive neuroscience. Despite the strengths of the manuscript, there are several areas where improvements could be made.

We thank the reviewer for this summary and positive appraisal of our study and our findings. In addition, we are of course grateful for the excellent suggestions for improvements that we have embraced to further strengthen our article.

Microsaccades or Saccades?In the manuscript, the terms "microsaccades" and "saccades" are used interchangeably. For instance, "microsaccades" are mentioned in the keywords, whereas "saccades" appear in the results section. It is crucial to differentiate between these two concepts. Saccades are large, often deliberate eye movements used for scanning and shifting attention, while microsaccades are small, involuntary movements that maintain visual perception during fixation. The authors note the connection between microsaccades and attention, but it is not well-recognized that saccades are directly linked to attention. Despite the paradigm involving a fixation point, it remains unclear whether large eye movements (saccades) were removed from the analysis. The authors mention the relationship between microsaccades and attention but do not clarify whether large eye movements (saccades) were excluded from the analysis. If large eye movements were removed during data processing, this should be documented in the manuscript, including clear definitions of "microsaccades" and "saccades." If such trials were not removed, the contribution of large eye movements to the results should be shown, and an explanation provided as to why they should be considered.

We thank the reviewer for raising this relevant point. Before turning to this relevant distinction, we first wish to clarify how, for our main aim of tracking the dynamics of ‘re-orienting in working memory’, any spatial modulation in gaze – be it driven by micro- or macro-saccades – suits this purpose. Having made this explicit, we also fully agree that disambiguating the nature of the saccade bias during internal focusing has additional value.

Because it is notoriously challenging (or at least inherently arbitrary) to draw an absolute fixed boundary between macro- and microsaccades, we instead decided to adopt a two-stage approach to our analysis (building on prior studies from our lab, e.g., de Vries et al., 2023; Liu et al., 2023; Liu et al., 2022). In the first step, we analysed spatial biases in all detected saccades no matter their size (hence our labelling of them as “saccades” when describing these analyses). In a second step, we decomposed and visualized the saccade-rate effect as a function of saccade size in degrees. This second stage directly exposed the ‘nature’ of the saccade bias, as we visualized in Figure 2c (with time on the x axis, saccade size on the y axis, and the spatial modulation color coded). Because these visualizations directly address this major comment, we have now made these key set of results much clearer in our work (we agree that our original visualization of this key aspect of our data was suboptimal). In addition, we have added similar plot for the saccade data in the test-phase in Supplementary Figure S2b.

These complementary analyses show how the saccade bias (more toward than away saccades) is indeed predominantly driven by small saccades (hence are labelling as “micro-saccades” when interpreting our findings), and less so by larger saccades associated with looking back all the way to the location where the memory item had been presented at encoding (positioned at 6 degrees). This is important as it helps to arbitrate between fixational/micro-saccadic eye-movement biases (previously associated with covert and internal attention shifts; cf. de Vries et al., 2023; Engbert and Kliegl, 2003; Hafed and Clark, 2002; Liu et al., 2023; Liu et al., 2022) vs. larger eye movements back to the original locations of the item (previously associated with ‘looking at nothing’ during memory retrieval and imagery; cf. Brandt and Stark, 1997; Ferreira et al., 2008; Johansson and Johansson, 2014; Laeng et al., 2014; Martarelli and Mast, 2013; Spivey and Geng, 2001). By adopting this visualization, we can show this while preserving the richness of our data, and without having to a-priori set an (inherently arbitrary) threshold for classifying saccades as either “macro” or “micro”.

Having explained our rationale, we nevertheless agree with the reviewer that it is worth showing how our time course results hold up when only considering fixational eye movements below 2 visual degrees, which we consider “fixational” provided that our memory stimuli at encoding were presented at 6 visual degrees from central fixation. We show this in Supplementary Figure S1. As can be seen below, our main saccade bias results stay almost the same when restricting our analyses exclusively to fixational saccades within 2 degrees, both when considering our data after the retrocue (Supplementary Figure S1a) as well as after the memory test (Supplementary Figure S1b).

Because we agree this is important complementary data, we have now added this as supplementary figures. In addition, we have added the results to our article. We also point to these additional corroborating findings at key instances in our article:

Page 5 (Results)

“As in prior studies from our lab with similar experimental set-ups, internal attentional focusing was predominantly driven by fixational micro-saccades (small, involuntary eye-movements around current fixation). To reveal this in the current study, we decomposed and visualized the observed saccade-rate effect as a function of saccade size (Figure 2c), following the same procedure as we have adopted in other recent studies on this bias (de Vries et al., 2023; Liu et al., 2023; Liu et al., 2022). As shown in the saccade-size-over-time plots in Figure 2c, also in the current study, the difference between toward and away saccades (with red colours denoting more toward saccades) was predominantly driven by fixational saccades in the micro-saccades range (< 2°).”

“Moreover, as shown in Supplementary Figure S1a, complementary analyses show that our time course (saccade bias) results hold even when exclusively considering eye movements below 2 visual degrees that we defined as “fixational” provided that the memory items were presented 6 visual degrees from the fixation during encoding. This further corroborates that the bias observed during internal attentional focusing was predominantly driven by fixational micro-saccades rather than looking back to the encoded location of the memory items (cf. Johansson and Johansson, 2014; Richardson and Spivey, 2000; Spivey and Geng, 2001; Wynn et al., 2019).”

Page 7 (Results):

“As shown in the corresponding saccade-size-over-time plots in Supplementary Figure S2b, consistent with what we observed following the cue, the difference between toward and away saccades following the test was again predominantly driven by saccades in the fixational microsaccade range (< 2°), and the time course (saccade bias) results hold even when exclusively considering fixational eye movements below 2 visual degrees (Supplementary Figure S1b). Thus, just like mnemonic focusing after the cue, re-orienting after the memory test was also predominantly reflected in fixational micro-saccades, and not looking back at the original location of the memory items that were encoded at 6 degrees away from central fixation.”

Alpha Lateralization in Attentional Re-orientingIn the attentional orienting section of the results (Figure 2), the authors effectively present EEG alpha lateralization results with time-frequency plots and topographic maps. However, in the attentional reorienting section (Figure 3), these visualizations are absent. It is important to note that the time period in attentional orienting differs from attentional re-orienting, and consequently, the time-frequency plots and topographic maps may also differ. Therefore, it may be invalid to compute alpha lateralization without a clear alpha activity difference. The authors should consider including timefrequency plots and topographic maps for the attentional re-orienting period to validate their findings.

We thank the reviewer also for this constructive suggestion. The reason we did not expand on the time-frequency maps and topographies at the test-stage was the relative lack of alpha effects at the test stage (compared to the clearer alpha modulations after the retrocue). Nevertheless, we agree that including these data will increase transparency and the comprehensiveness of our article. We now added time-frequency plots and topographic maps for alpha lateralization in response to the workingmemory test in Supplementary Figure S2. As can be seen, the time-frequency plots and topographies in the re-focusing period after the working-memory test were consistent with our time-series plots in Figure 3a – reinforcing how alpha lateralization is generally not clear following the working-memory test. In accordance with this relevant addition, we added the following in the revised manuscript:

Page 7 (Results):

“For complementary time-frequency and topographical visualizations, see Supplementary Figure S2a.”

Onset and Offset Latency of Saccade BiasThe use of the 50% peak to determine the onset and offset latency of the saccade bias is problematic. For example, if one condition has a higher peak amplitude than another, the standard for saccade bias onset would be higher, making the observed differences between the onset/offset latencies potentially driven by amplitude rather than the latencies themselves. The authors should consider a more robust method for determining saccade bias onset and offset that accounts for these amplitude differences.

We thank the reviewer for raising this valuable point. We agree that the calculation of onset and offset latencies of the saccade bias could be influenced by the peak amplitude of the waveforms. Thus, we further conducted the Fractional Area Latency (FAL) analysis on the comparison of the saccade bias following the working-memory test between valid cue (expected test) and invalid cue (unexpected test) trials. The FAL analysis has been commonly applied to Event-Related Potentials (ERPs) to estimate the latency of ERP components (Hansen and Hillyard, 1980; Luck, 2005). Instead of relying on the peak latency, the FAL method calculates latency based on a predefined fraction of the area under the waveform. This can provide a more robust measure of component latency. Prompted by this comment, we now also applied FAL analysis to our saccade bias waveforms. This corroborated our original conclusion. Because we believe this is an important complement, we now added these additional outcomes to our article:

Page 9 (Results):

“We additionally conducted Fractional Area Latency (FAL) analysis on the comparison of the saccade bias following the memory test between valid- and invalid-cue trials to rule out the potential contribution of peak amplitude differences into the onset and offset latency differences (Hansen and Hillyard, 1980; Kiesel et al., 2008; Luck, 2005). Consistent with our jackknife-based latency analysis, the FAL analysis revealed a significantly prolonged saccade bias following the unexpected tests (the invalid-cue trials) vs. expected tests (the valid-cue trials) in both 80% and 60% cue-reliability conditions (411 ms vs. 463 ms, *t*_(14)_ = 2.358, *p* = 0.034; 417 ms vs. 468 ms, *t*_(15)_ = 2.168, *p* = 0.047; for 80% and 60%, respectively). Again, there was no significant difference in onset latency following unexpected vs. expected tests. (346 ms vs. 374 ms, *t*_(14)_ = 2.052, *p* = 0.060; 353 ms vs. 401 ms, *t*_(15)_ = 1.577, *p* = 0.136; for 80% and 60%, respectively).”

In accordance, we also added the following to our Methods:

Page 18 (Methods):

“In addition to the jackknife-based latency analysis, we further applied a Fractional Area Latency (FAL) method to the saccade bias comparison between validly and invalidly cued memory tests to rule out the contribution of the peak amplitude difference into the onset and offset latency difference (Hansen and Hillyard, 1980; Kiesel et al., 2008; Luck, 2005). We first defined the onset and offset latency of the saccade bias as the first time point at which 25% or 75% of the total area of the component has been reached, relative to a lower boundary of a difference of 0.3 Hz between toward and away saccades (to remove the influence of noise fluctuations in our difference time course below this lower boundary). The extracted onset and offset latency for all participants was then compared using paired-samples t-tests.”

Control Analysis for Trials Not Using the Initial CueThe control analysis for trials where participants did not use the initial cue raises several questions:(1) The authors claim that "unlike continuous alpha activity, saccades are events that can be classified on a single-trial level." However, alpha activity can also be analyzed at the single-trial level, as demonstrated by studies like "Alpha Oscillations in the Human Brain Implement Distractor Suppression Independent of Target Selection" by Wöstmann et al. (2019). If single-trial alpha activity can be used, it should be included in additional control analyses.

We agree with the reviewer that alpha activity can also be analyzed at the single-trial level. However, because alpha is a continuous signal, single-trial alpha activity will necessarily be graded (trials with *more* or *less* alpha power). This is still different from saccades, that are not continuous signals but true ‘events’ (either a saccade was made, or no saccade was made, with no continuum in between). Because of this unique property, it is possible to sort trials by whether a saccade was present (and, if present, by its direction), in an all-or-none way that is not possible for alpha activity that can only be sorted by its graded amplitude/power. This is the key distinction underlying our motivation to sort the trials based on saccades, as we now make clearer:

Page 10 (Results):

“Although alpha can also be analyzed as the single trial level (e.g. Macdonald et al., 2011; Wöstmann et al., 2019; for a review, see Kosciessa et al., 2020), saccades offer the unique opportunity to split trials not by graded amplitude fluctuations but by discrete all-or-none events.”

In addition, please note how our saccade markers were also more reliable/sensitive, especially in the subsequent memory-test-phase of interest. This is another reason we decided to focus this control analysis on saccades and not alpha activity.

(2) The authors aimed to test whether the re-orienting signal observed after the test is not driven exclusively by trials where participants did not use the initial cue. They hypothesized that "in such a scenario, we should only observe attention deployment after the test stimulus in trials in which participants did not use the preceding retro cue." However, if the saccade bias is the index for attentional deployment, the authors should conduct a statistical test for significant saccade bias rather than only comparing toward-saccade after-cue trials with no-toward-saccade after-cue trials. The null results between the two conditions do not immediately suggest that there is attention deployment in both conditions.

We thank the reviewer for bringing up this important point. We fully agree and, in fact, we had conducted the relevant statistical analysis for each of the conditions separately (in addition to their comparison). Upon reflection, we came to realize that in our original submission it was easy to overlook this point, and therefore thank the reviewer for flagging this. To make this clearer, we now also added the relevant statistical clusters in Figure 4a,b and more clearly report them in the associated text:

Page 10 (Results):

“As we show in Figure 4a,b, we found clear gaze signatures of attentional deployment in response to expected (valid) memory tests, no matter whether we had pre-selected trials in which we had also seen such deployment after the cue in gaze (cluster P: 0.115, 0.041, 0.027, <0.001 for 80%-valid, 60%-valid, 80%-invalid, 60%-invalid trials, respectively), or not (cluster P: 0.016, 0.009, 0.001, <0.001 for 80%-valid, 60%-valid, 80%-invalid, 60%-invalid trials, respectively).”

(3) Even if attention deployment occurs in both conditions, the prolonged re-orienting effect could also be caused by trials where participants did not use the initial cue. Unexpected trials usually involve larger and longer brain activity. The authors should perform the same analysis on the time after the removal of trials without toward-saccade after the cue to address this potential confound.

We thank the reviewer for raising this. It is crucial to point out, however, that after any given 80% or 60% reliable cue, the participants cannot yet know whether the subsequent memory test in that trial will be expected (valid cue) or unexpected (invalid cue). Accordingly, the prolonged re-orienting after unexpected vs. expected memory tests cannot be explained by differential use of the cue (i.e., differential cue-use cannot be a “confound” for differential responses to expected and unexpected memory tests, as observed within the 80 and 60% cue-reliability conditions).

**Reviewer #2 (Public Review):**
Summary:This study utilized EEG-alpha activity and saccade bias to quantify the spatial allocation of attention during a working memory task. The findings indicate a second stage of internal attentional deployment following the appearance of a memory test, revealing distinct patterns between expected and unexpected test trials. The spatial bias observed during the expected test suggests a memory verification process, whereas the prolonged spatial bias during the unexpected test suggests a reorienting response to the memory test. This work offers novel insights into the dynamics of attentional deployment, particularly in terms of orienting and re-orienting following both the cue and memory test.Strengths:The inclusion of both EEG-alpha activity and saccade bias yields consistent results in quantifying the attentional orienting and re-orienting processes. The data clearly delineate the dynamics of spatial attentional shifts in working memory. The findings of a second stage of attentional re-orienting may enhance our understanding of how memorized information is retrieved.Weaknesses:Although analyses of neural signatures and saccade bias provided clear evidence regarding the dynamics of spatial attention, the link between these signatures and behavioral performance remains unclear. Given the novelty of this study in proposing a second stage of 'verification' of memory contents, it would be more informative to present evidence demonstrating how this verification process enhances memory performance.

We thank the reviewer for the positive summary of our work and for highlighting key strengths. We also appreciate the constructive suggestions, such as addressing the link between our observed refocusing signals and behavioral performance in our task. We now performed these additional analyses and added their outcomes to the revised article, as we detail in response to comment 2 below.

**Reviewer #2 (Recommendations For The Authors):**
(1) Figure 2 shows graded spatial modulations in both EEG-alpha activity and saccade bias. However, while the imperative 100% cue conditions and 100% validity conditions largely overlap in EEG-alpha activity, a clear difference is present between these two conditions in saccade bias. The cause of the difference in saccade bias is unclear.

We thank the reviewer for pointing out this interesting difference. At this stage, it is hard to know with certainty whether this reflects a genuine difference in our 100% reliable and 100% imperative cue conditions that is selectively picked up by our gaze but not alpha marker. Alternatively, this may reflect differential sensitivity of our two markers to different sources of noise. Either way, we agree that this observation is worth calling out and reflecting on when discussing these results:

Page 6 (Results):

“It’s worth noting that while alpha lateralization shows very comparable amplitudes in the imperative-100% and 100% conditions, the saccade bias was larger following imperative-100% vs. 100% reliable cues. This may reflect a difference between these two cueing conditions that is selectively picked up by our gaze marker (though it may also reflect differential sensitivity of our two markers to different sources of noise). […]”

(2) Figure 3 shows signatures of attentional re-orienting after the memory test presented at the center. When the cue was not 100% valid, a noticeable saccade bias towards the memorized location of the test item was observed. This finding was explained as reflecting a re-orienting to the mnemonic contents. To strengthen this interpretation, I suggest providing evidence for the link between the attentional re-orienting signatures and memory performance.

We thank the reviewer for this constructive suggestion. We now sorted trials by behavioral performance using a median split on RT (fast-RT vs. slow-RT trials) and reproduction error (highaccuracy vs. low-accuracy trials). Because we believe the outcomes of these analyses increase transparency as well as the comprehensiveness of our article, we have now included them as Supplementary Figure S3.

As shown below, we were able to link the saccade bias following the memory test to subsequent performance, but this reached significance only for the 80% valid-cue trials when splitting by RT (cluster P = 0.001). For the other conditions, we could not establish a reliable difference by our performance splits. Possibly this is due to a lack of sensitivity, given the relatively large number of conditions we had and, consequently, the relatively small number of trials we therefore had per condition (particularly in the invalid-cue condition with unexpected memory tests). We now bring forward these additional outcomes at the relevant section in our Results:

Page 7 (Results):

“We also sorted patterns of gaze bias after the memory test by performance but could only establish a link between this gaze bias and RT following expected memory tests in our 80% cuereliability condition (cluster P = 0.001, Supplementary Figure S3). The lack of significant statistical differences in the remaining conditions may possibly reflect a lack of sensitivity (insufficient trial numbers) for this additional analysis.”

(3) When comparing the time course of attentional re-orienting after the memory test, a prolonged attentional re-orienting was observed for unexpected memory tests compared to the expected ones. While the onset latency was similar for unexpected and expected memory tests, the offset latency was prolonged for the unexpected memory test. Could this be attributed to the learned tendency to saccade toward the expected location in more valid trials? In this case, the prolonged re-orienting may indicate increased efforts in suppressing the previously learned tendency.

We thank the reviewer for bringing up this interesting possibility. In our original interpretation, this prolonged signal reflects a longer time needed to bring the unexpected memory content ‘back in focus’ before being able to report its orientation. At the same time, we agree that there are alternative explanations possible, such as the one raised by the reviewer. We now make this clearer when discussing this finding:

Page 14 (Discussion):

“[…] attentional deployment did become prolonged when re-focusing the unexpected memory item, likely reflecting prolonged effort to extract the relevant information from the memory item that was not expected to be tested. However, there may also be alternative accounts for this observation, such as suppressing a learned tendency to saccade in the direction of the expected item following an unexpected memory test.”

(4) To test whether the re-orienting signature is predominantly influenced by trials where participants delayed the use of cue information until the memory test appeared, the authors sorted the trials based on saccade bias after the initial cue. However, it would be more informative to depict the reorienting patterns by sorting trials based on memory performance. The rationale is that in trials where participants delayed using the initial retro-cue, memory performance (e.g., measured by reproduction error) might be less precise due to the extended memory retention period. Compared to saccade bias for initial orienting, memory performance could provide more reliable evidence as it represents a more independent measure.

We thank the reviewer for this suggestion. As delineated in response to comment 2, we now conducted this additional analysis and added the relevant outcomes to our article.

(5) While the number of trials was well-balanced across blocks (~ 240 trials), how did the authors address the imbalance between valid and invalid trials, especially in the 80% cue validity block?

We thank the reviewer for raising this point. First, we wish to point out that while trial numbers will indeed impact the sensitivity for finding an effect, trial numbers do not bias the mean – and therefore also not the comparison between means. In this light, it is vital to appreciate that our findings do not reflect a significant effect in valid trials but no significant effect in invalid trials (which we agree could be due to a difference in trial numbers), but rather a statistical difference between valid and invalid trials. This significant difference in the means between valid and invalid true cannot be attributed to a difference in trial numbers between these conditions.

Having clarified this, we nevertheless agree that it is also worthwhile to empirically validate this assertion and show how our findings hold even when carefully matching the number of trials between valid and invalid conditions (i.e., between expected and unexpected memory tests). To do so, we ran a sub-sampling analysis where we sub-sampled the number of valid trials to match the number of invalid trials available per condition (and averaged the results across 1000 random sub-samplings to increase reliability). As anticipated, this replicated our findings of robust differences between the gaze bias following expected and unexpected memory tests in both our 80 and 60% cue-reliability conditions. We now present these additional outcomes in Supplementary Figure S4.

Because we agree this is an important re-assuring control analysis, we have now added this to our article:

Page 9 (Results):

“To rule out the possibility that the saccade-bias differences following expected and unexpected memory tests are caused by uneven trial numbers (200 vs. 50 trials in the 80% cuereliability condition, 150 vs. 100 trials in the 60% cue-reliability condition), we ran a subsampling analysis where we sub-sampled the number of valid trials to match the number of invalid trials available per condition (averaging the results across 1000 random sub-samplings to increase reliability). As shown in Supplementary Figure S4, this complementary subsampling analysis confirmed that our observed differences between the saccade bias following expected and unexpected memory tests in both 80% and 60% cue-reliability conditions are robust even when carefully matching the number of trials between validly cued (expected) and invalidly cued (unexpected) memory test.”

**Reviewer #3 (Public Review):**
Summary:Wang and van Ede investigate whether and how attention re-orients within visual working memory following expected and unexpected centrally presented memory tests. Using a combination of spatial modulations in neural activity (EEG-alpha lateralization) and gaze bias quantified as time courses of microsaccade rate, the authors examined how retro cues with varying levels of reliability influence attentional deployment and subsequent memory performance. The conclusion is that attentional reorienting occurs within visual working memory, even when tested centrally, with distinct patterns following expected and unexpected tests. The findings provide new value for the field and are likely of broad interest and impact, by highlighting working memory as an action-bound process (in)dependent on (an ambiguous) past.Strengths:The study uniquely integrates behavioral data (accuracy and reaction time), EEG-alpha activity, and gaze tracking to provide a comprehensive analysis of attentional re-orienting within visual working memory. As typical for this research group, the validity of the findings follows from the task design that effectively manipulates the reliability of retro cues and isolates attentional processes related to memory tests. The use of well-established markers for spatial attention (i.e. alpha lateralization) and more recently entangled dependent variable (gaze bias) is commendable. Utilizing these dependent metrics, the concise report presents a thorough analysis of the scaling effects of cue reliability on attentional deployment, both at the behavioral and neural levels. The clear demonstration of prolonged attentional deployment following unexpected memory tests is particularly noteworthy, although there are no significant time clusters per definition as time isn't a factor in a statistical sense, the jackknife approach is convincing. Overall, the evidence is compelling allowing the conclusion of a second stage of internal attentional deployment following both expected and unexpected memory tests, highlighting the importance of memory verification and re-orienting processes.Weaknesses:I want to stress upfront that these weaknesses are not specific to the presented work and do not affect my recommendation of the paper in its present form.The sample size is consistent with previous studies, a larger sample could enhance the generalizability and robustness of the findings. The authors acknowledge high noise levels in EEG-alpha activity, which may affect the reliability of this marker. This is a general issue in non-invasive electrophysiology that cannot be handled by the authors but an interested reader should be aware of it. Effectively, the sensitivity of the gaze analysis appears "better" in part due to the better SNR. The latter also sets the boundaries for single-tiral analyses as the authors correctly mention. In terms of generalizability, I am convinced that the main outcome will likely generalize to different samples and stimulus types. Yet, as typical for the field future research could explore different contexts and task demands to validate and extend the findings. The authors provide here how and why (including sharing of data and code).

We thank the reviewer for summarising our work and for carefully delineating its strengths. We also appreciate the mentioning of relevant generic limitations and agree that important avenues for future studies will be to expand this work with larger sample sizes, complementary measurement techniques, and complementary task contexts and stimuli.

**Reviewer #3 (Recommendations For The Authors):**
In the conclusion, Wang and van Ede successfully demonstrate that attentional re-orienting occurs within visual working memory following both expected and unexpected tests. The conclusions are supported by the data and analyses applied, showing that attentional deployment is by the reliability of retro cues. Centrally presented memory tests can invoke either a verification or a revision of internal focus, the latter thus far not considered in both theory and experimental design in cognitive neuroscience.I don't have any recommendations that will significantly change the conclusions.

We thank the reviewer for having carefully evaluated our work and hope the reviewer will also perceive the changes we made and the additional analyses we added in responses to the other two reviewers as further strengthening our article.

Reference

Brandt SA, Stark LW. 1997. Spontaneous eye movements during visual imagery reflect the content of the visual scene. *J Cogn Neurosci* 9. doi:10.1162/jocn.1997.9.1.27

de Vries E, Fejer G, van Ede F. 2023. No obligatory trade-off between the use of space and time for working memory. *Communications Psychology*.

Engbert R, Kliegl R. 2003. Microsaccades uncover the orientation of covert attention. *Vision Res* 43. doi:10.1016/S0042-6989(03)00084-1

Ferreira F, Apel J, Henderson JM. 2008. Taking a new look at looking at nothing. *Trends Cogn Sci* 12. doi:10.1016/j.tics.2008.07.007

Hafed ZM, Clark JJ. 2002. Microsaccades as an overt measure of covert attention shifts. *Vision Res* 42. doi:10.1016/S0042-6989(02)00263-8

Hansen JC, Hillyard SA. 1980. Endogeneous brain potentials associated with selective auditory attention. *Electroencephalogr Clin Neurophysiol* 49. doi:10.1016/0013-4694(80)90222-9

Johansson R, Johansson M. 2014. Look Here, Eye Movements Play a Functional Role in Memory Retrieval. *Psychol Sci* 25. doi:10.1177/0956797613498260

Kiesel A, Miller J, Jolicœur P, Brisson B. 2008. Measurement of ERP latency differences: A comparison of single-participant and jackknife-based scoring methods. *Psychophysiology* 45. doi:10.1111/j.1469-8986.2007.00618.x

Kosciessa JQ, Grandy TH, Garrett DD, Werkle-Bergner M. 2020. Single-trial characterization of neural rhythms: Potential and challenges. *Neuroimage* 206. doi:10.1016/j.neuroimage.2019.116331

Laeng B, Bloem IM, D’Ascenzo S, Tommasi L. 2014. Scrutinizing visual images: The role of gaze in mental imagery and memory. *Cognition* 131. doi:10.1016/j.cognition.2014.01.003

Liu B, Alexopoulou SZ, van Ede F. 2023. Jointly looking to the past and the future in visual working memory. *Elife*.

Liu B, Nobre AC, van Ede F. 2022. Functional but not obligatory link between microsaccades and neural modulation by covert spatial attention. *Nat Commun* 13. doi:10.1038/s41467-022-312173

Luck S. 2005. Ten Simple Rules for Deisgning ERP Experiments. *Event-related potentials: A methods handbook*.

Macdonald JSP, Mathan S, Yeung N. 2011. Trial-by-trial variations in subjective attentional state are reflected in ongoing prestimulus EEG alpha oscillations. *Front Psychol* 2. doi:10.3389/fpsyg.2011.00082

Martarelli CS, Mast FW. 2013. Eye movements during long-term pictorial recall. *Psychol Res* 77. doi:10.1007/s00426-012-0439-7

Richardson DC, Spivey MJ. 2000. Representation, space and Hollywood Squares: Looking at things that aren’t there anymore. *Cognition* 76. doi:10.1016/S0010-0277(00)00084-6

Spivey MJ, Geng JJ. 2001. Oculomotor mechanisms activated by imagery and memory: Eye movements to absent objects. *Psychol Res* 65. doi:10.1007/s004260100059

van Ede F, Chekroud SR, Nobre AC. 2019. Human gaze tracks attentional focusing in memorized visual space. *Nat Hum Behav*. doi:10.1038/s41562-019-0549-y

Wöstmann M, Alavash M, Obleser J. 2019. Alpha oscillations in the human brain implement distractor suppression independent of target selection. *Journal of Neuroscience* 39. doi:10.1523/JNEUROSCI.1954-19.2019

Wynn JS, Shen K, Ryan JD. 2019. Eye movements actively reinstate spatiotemporal mnemonic content. *Vision (Switzerland)* 3. doi:10.3390/vision3020021